# Mitotic spindle association of TACC3 requires Aurora-A-dependent stabilization of a cryptic α-helix

Selena G Burgess[1,†], Manjeet Mukherjee[1,†], Sarah Sabir[1], Nimesh Joseph[2], Cristina Gutiérrez-Caballero[3], Mark W Richards[1], Nicolas Huguenin-Dezot[4], Jason W Chin[4], Eileen J Kennedy[5], Mark Pfuhl[6] [ID], Stephen J Royle[3] [ID], Fanni Gergely[2] [ID] & Richard Bayliss[1,*] [ID]

## Abstract

Aurora-A regulates the recruitment of TACC3 to the mitotic spindle through a phospho-dependent interaction with clathrin heavy chain (CHC). Here, we describe the structural basis of these interactions, mediated by three motifs in a disordered region of TACC3. A hydrophobic docking motif binds to a previously uncharacterized pocket on Aurora-A that is blocked in most kinases. Abrogation of the docking motif causes a delay in late mitosis, consistent with the cellular distribution of Aurora-A complexes. Phosphorylation of Ser558 engages a conformational switch in a second motif from a disordered state, needed to bind the kinase active site, into a helical conformation. The helix extends into a third, adjacent motif that is recognized by a helical-repeat region of CHC, not a recognized phospho-reader domain. This potentially widespread mechanism of phospho-recognition provides greater flexibility to tune the molecular details of the interaction than canonical recognition motifs that are dominated by phosphate binding.

**Keywords** disorder–order transition; intrinsically disordered protein; phosphorylation; protein kinase; protein–protein interaction
**Subject Categories** Cell Adhesion, Polarity & Cytoskeleton; Cell Cycle; Structural Biology
**The EMBO Journal (2018) 37: e97902**

## Introduction

Aurora-A is a Ser/Thr protein kinase that regulates mitotic entry and mitotic spindle assembly (Nikonova *et al*, 2013) through phosphorylation of many substrates. These include TACC3, a member of the transforming acidic coiled-coil family of proteins, which have conserved functions in microtubule dynamics (Peset & Vernos, 2008). The requirement of Aurora-A phosphorylation for localization of TACC3 to spindle microtubules is conserved in the *Drosophila* ortholog D-TACC, and the *Xenopus* ortholog maskin (Giet *et al*, 2002; Kinoshita *et al*, 2005; Peset *et al*, 2005). Phosphorylation of human TACC3 on S558 drives an interaction with the ankle region of CHC to form a complex that binds the microtubule lattice through the coiled-coil region of TACC3 and the β-propeller domain of CHC (Fig 1A, right; Fu *et al*, 2010; Hubner *et al*, 2010; Lin *et al*, 2010; Hood *et al*, 2013). This complex, together with ch-TOG, localizes along the spindle forming bridges between parallel microtubules of the kinetochore fibres (K-fibres). These bridges help to stabilize K-fibres and contribute to spindle robustness while unphosphorylated TACC3 is targeted to the plus-ends of microtubules by ch-TOG independently of Aurora-A and CHC (Burgess *et al*, 2015; Gutierrez-Caballero *et al*, 2015). CHC lacks a canonical phospho-reader domain and it is unclear how it recognizes phosphorylated TACC3.

The activity of Aurora-A is controlled through phosphorylation of its activation loop on T288 (Bayliss *et al*, 2003; Eyers *et al*, 2003). Like many of its other binding partners, such as TPX2, I-2, HEF1, and N-Myc, TACC3 is an Aurora-A activator (Nikonova *et al*, 2013; Richards *et al*, 2016). However, only in the case of TPX2 has the mechanism of activation, and how this synergizes with phosphorylation of T288, been resolved (Bayliss *et al*, 2003; Eyers *et al*, 2003; Dodson & Bayliss, 2012). Amino acids (aa) 1–43 of TPX2 stimulate Aurora-A activity, whether T288 is phosphorylated or not, and

1   Astbury Centre for Structural Molecular Biology, School of Molecular and Cellular Biology, Faculty of Biological Sciences, University of Leeds, Leeds, UK
2   Cancer Research UK Cambridge Institute, Li Ka Shing Centre, University of Cambridge, Cambridge, UK
3   Centre for Mechanochemical Cell Biology, Warwick Medical School, University of Warwick, Coventry, UK
4   Medical Research Council Laboratory of Molecular Biology, Cambridge, UK
5   Department of Pharmaceutical and Biomedical Sciences, College of Pharmacy, University of Georgia, Athens, GA, USA
6   Cardiovascular & Randall Division, Kings College London, London, UK
    *Corresponding author. Tel: +44 113 3439919; E-mail: r.w.bayliss@leeds.ac.uk
    †These authors contributed equally to this work

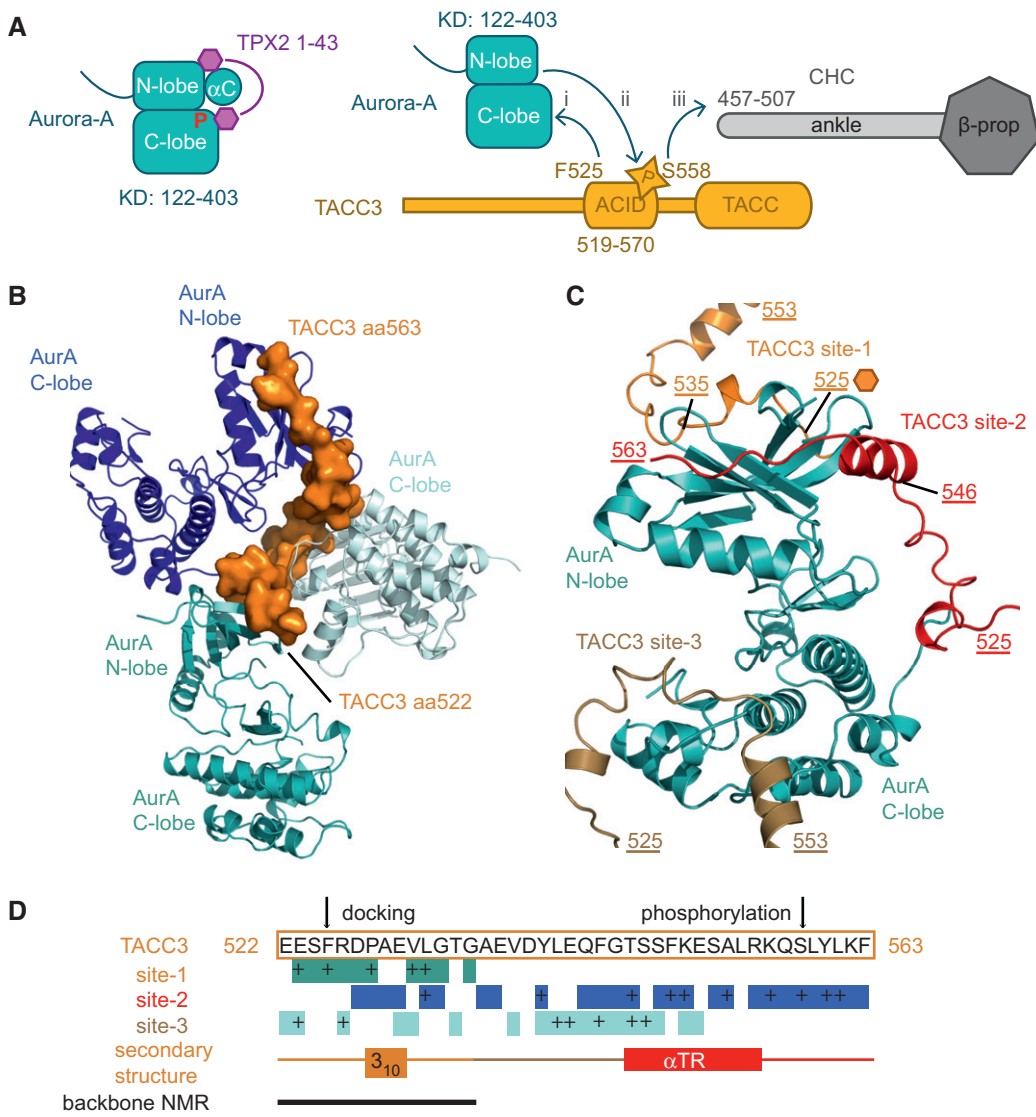

**Figure 1.  Three interaction sites in the crystal structure of Aurora-A/TACC3.**

A   Schematic illustration showing the domain structures and interacting regions of Aurora-A, TPX2, CHC and TACC3. TPX2 1–43 stabilizes the active conformation of Aurora-A through interactions with the αC-helix and the phosphate-bearing activation loop (left). TACC3 F525 is crucial for the interaction with Aurora-A (i). Phosphorylation of TACC3 S558 by Aurora-A (ii) is required for the subsequent interaction with the CHC ankle region (iii).

B   View of the Aurora-A/TACC3 structure centred on the TACC3 molecule (orange surface), which interacts with three molecules of Aurora-A (blue, light blue, teal cartoon).

C   View of the Aurora-A/TACC3 structure centred on one Aurora-A molecule (teal cartoon), which interacts with three molecules of TACC3 (orange, red, brown).

D   Diagram showing which TACC3 residues interact with Aurora-A at each of the three sites (teal, blue and light blue boxes), with key interactions marked as "+" and secondary structure of TACC3 indicated. Backbone chemical shift mapping is of inactive Aurora-A upon addition of TACC3$^{N-ACID}$ (solid line; Burgess et al, 2015).

TPX2-binding and T288 phosphorylation together stabilize the αC-helix and activation loop in a fully ordered conformation (Fig 1A, left) similar to that observed in active Ser/Thr kinases, such as PKA (Bayliss et al, 2003; Goldsmith et al, 2007; Dodson & Bayliss, 2012; Zorba et al, 2014; Cyphers et al, 2017). This mechanism is antagonized by a single domain antibody that traps Aurora-A into an inactive conformation (Burgess et al, 2016). The only other structure of a physiological Aurora-A protein complex is with a fragment of N-Myc, which binds primarily to the activation loop and only partially overlaps the TPX2 binding site (Richards et al, 2016). TACC3 binds Aurora-A through a region 30aa N-terminal to the S558 phosphorylation site, centred on a critical phenylalanine residue, F525 (Burgess et al, 2015). The location of the TACC3 binding site on Aurora-A is unknown, but is distinct from that of TPX2 because a 3-way complex can be formed in vitro (Burgess et al, 2015). It is not known whether Aurora-A/TPX2 and Aurora-A/TACC3 complexes co-localize on the mitotic spindle, or how their localizations change as mitosis develops.

Aurora-A regulation of binding between TACC3 and CHC is critical for the proper and timely assembly of the mitotic spindle. We have mapped the interacting regions that mediate the processes by which TACC3 is recognized by Aurora-A, phosphorylated, and then handed over to CHC, but detailed structural insights are required to determine how TACC3 activates Aurora-A without binding to the same regulatory site as TPX2 and how phosphorylation of TACC3 directs its interaction with CHC. Here, we find that the region around F525 of TACC3 is a docking motif that binds to a site on the surface of Aurora-A that is blocked in kinases such as PKA. Disruption of the docking motif by point mutation impairs localization of TACC3 on the mitotic spindle and delays the completion of mitosis. In unphosphorylated TACC3, the region between the docking site and S558 contains a nascent α-helix. Phosphorylation at S558 stabilizes and extends this α-helix, displaying a pattern of hydrophobic residues that is recognized by a complementary surface in the ankle region of CHC.

Our study extends the current paradigm that the role of phosphorylation in signalling pathways is to promote complex formation via phospho-specific reader modules because a phosphorylation-induced helix could be read by a wide range of proteins, most obviously those comprising helical repeats like CHC. Furthermore, this mechanism of phosopho-recognition, in which the sequence motif that governs the stabilization of the helix is distinct from that used to form the interface with the reader, provides greater flexibility to tune the molecular details of the interaction than canonical recognition motifs which are dominated by phosphate binding.

# Results

### Crystal structure of the Aurora-A$^{KD}$/TACC3$^{N-ACID}$ complex

Our previous work, summarized in Fig 1A, mapped the regions that interact in the complex between the Aurora-A kinase domain and the N-terminal fragment of the Aurora-A and CHC interaction domain (ACID) of TACC3 corresponding to aa519–563 (TACC3$^{N-ACID}$; Fig EV1A; Burgess *et al*, 2015).

Crystals of TACC3$^{N-ACID}$ bound to an Aurora-A 122–403 C290A, C393A, D274N mutant (Aurora-A$^{M3KD}$) diffracted to 2.0 Å resolution (Table 1). The structure was solved by molecular replacement (MR) using an existing Aurora-A structure (PDB 4CEG; Burgess & Bayliss, 2015). Unambiguous electron density for TACC3$^{N-ACID}$ was observed (Fig EV1B), and one complex was located in the asymmetric unit. Residues 126–280 and 290–392 of Aurora-A$^{M3KD}$ were included in the final model, with the part of the activation loop being disordered consistent with its unphosphorylated state (Cheetham *et al*, 2002; Burgess *et al*, 2016). TACC3$^{N-ACID}$ mostly adopts an extended conformation but with two short regions of secondary structure: a $3_{10}$-helix formed by 528–530, and an α-helix by 546–555 (αTR). Due to crystal packing, each molecule of TACC3$^{N-ACID}$ contacts three molecules of Aurora-A$^{M3KD}$ (Fig 1B). These contacts cover the entire length of TACC3$^{N-ACID}$, which allowed it to be fully resolved, and involved three distinct sites on the surface of Aurora-A$^{M3KD}$ (labelled as sites 1–3; Fig 1C and D). However, this is unlikely to represent the physiological arrangement of the complex and it would not be possible for a single TACC3$^{N-ACID}$ molecule to occupy all three sites simultaneously (Fig 1D).

Of the three sites, only the contact at site 1 is consistent with our previous data: an F525A mutation increased the $K_d$ of the interaction from 5.7 to >150 μM and chemical shift changes in $^1$H-$^{15}$N HSQC spectra of TACC3$^{N-ACID}$ upon addition of Aurora-A were restricted to aa521–535 (solid black line, Fig 1D; Burgess *et al*, 2015). Interestingly, site 2 overlaps with the binding site for TPX2, but there is no conserved secondary structure between TPX2 and TACC3 and the chains run in opposite directions over the surface of Aurora-A. To further validate in solution the conformation and interactions of TACC3 519–540 observed in the crystal structure, we extended our previous nuclear magnetic resonance (NMR) studies by fully assigning C and H atom resonances in the 519–540 region both free in solution and bound to Aurora-A (Fig EV1C and D). The greatest observed changes in chemical shift were observed for H atoms in side chains that contact Aurora-A site 1 in the crystal structure such as F525 (Hδ and Hε) and P528 (Hδ; Fig EV1C) and the side chain of R526 (Hδ), which makes only modest contact with Aurora-A but folds back against the $3_{10}$ helix of TACC3, close to the acidic side chain of E530 (Fig 2A). Also, upon binding, the chemical shifts of the L532 Hδ atoms are clearly shifted away from their positions in the free form, one upfield and one downfield, by packing against Y148 of Aurora-A (Figs 2A and B, and EV1D). The changes in chemical shifts are therefore consistent with conformational changes in TACC3 and interactions with Aurora-A at site 1.

### Mutations at site 1 disrupt the interaction and reduce Aurora-A activation

The interaction at site 1 involves residues 523–535 of TACC3$^{N-ACID}$ binding to the surface of the β-sheet in the Aurora-A N-lobe (Fig 2A and B) contacting β-strands β1–β3 and β5 and adopting an extended conformation, with the exception of a short $3_{10}$-helix at aa527–532. The total surface area buried in this interface is ~950 Å$^2$, and the key contributions are made by TACC3 residues F525, L532, P528, and Aurora-A residues I135 and R151 (Fig 2B). I135 of Aurora-A is sandwiched between the side chains of TACC3 F525 and L532. The F525 side chain is inserted into a pocket formed by the side chains of R151, I158, L149, E134, and main chains of G136 and A150, of Aurora-A. The interface also involves an intermolecular salt bridge between E523 of TACC3 and R137 of Aurora-A and a H-bond between the main chain oxygen of TACC3 R526 and the main chain amine of Aurora-A I135.

The contributions of the site 1 residues to the interaction were quantified using fluorescence polarization spectroscopy (FP) with a set of fluorescein (FAM)-labelled peptides. The $K_d$ of FAM-WT TACC3 519–570 binding to Aurora-A was measured as 3.0 ± 0.2 μM (Fig EV2A), which is in good agreement with the $K_d$ measured by microscale thermophoresis (MST) and NMR for TACC3$^{N-ACID}$ binding to Aurora-A (6–8 μM; Burgess *et al*, 2015). The $K_d$ of the WT TACC3 522–536 peptide binding to Aurora-A was 20 ± 1 μM by FP suggesting that the docking interaction at site 1 is the major contributor to the overall interaction while the region of TACC3 in the vicinity of S558 makes a minor contribution, consistent with an enzyme–substrate interaction (Fig 2C and D; Burgess *et al*, 2015). The affinity of a TACC3 522–536 peptide incorporating the F525A mutation was extremely weak (2.3 ± 0.4 mM), consistent with the major contribution of F525 to the interface (155 Å$^2$ buried surface area). L532 contributes

**Table 1. Data collection and refinement statistics.**

| | Aurora-A[M3KD]/TACC3 519–563 (PDB 5ODT) | CHC-(TGS)$_4$TACC3 549–570[pS558] (PDB 5ODS) |
|---|---|---|
| **Data collection** | | |
| Space group | *I*2 3 | P 1 2$_1$ 1 |
| Cell dimensions | | |
| *a*, *b*, *c* (Å) | 137.04, 137.04, 137.04 | 115.52, 120.04, 123.13 |
| α, β, γ (°) | 90.00, 90.00, 90.00 | 90.00, 95.72, 90.00 |
| Resolution range (Å) | 68.52–2.02 (2.07–2.02)[a] | 61.26–3.09 (3.14–3.09)[a] |
| $R_{merge}$ (%) | 8.7 (193) | 8.4 (123) |
| *I*/σ*I* | 25.9 (1.9) | 10.3 (1.2) |
| Completeness (%) | 100 (100) | 98.7 (99.6) |
| Redundancy | 20.2 (20.5) | 3.3 (3.4) |
| **Refinement** | | |
| Resolution (Å) | 68.52–2.02 | 61.26–3.09 |
| No. reflections | 28,159 | 60,649 |
| $R_{work}$/$R_{free}$ | 17.78/21.11 | 22.08/26.9 |
| No. atoms | | |
| Protein | 2,372 | 18,224 |
| Water | 140 | 0 |
| Hetero | 65 | 0 |
| Mean *B*-factors | | |
| Protein | 43.78 | 124.5 |
| Hetero | 57.19 | – |
| Water | 49.84 | – |
| Wilson *B*-factor | 41.27 | 99.33 |
| r.m.s. deviations | | |
| Bond lengths (Å) | 0.008 | 0.006 |
| Bond angles (°) | 1.047 | 1.204 |
| **MolProbity analysis** | | |
| All-atom clash-score | 3.35 | 5.96 |
| Ramachandran allowed (%) | 3.05 | 5.06 |
| Ramachandran outliers (%) | 0 | 0 |
| Ramachandran favoured (%) | 96.95 | 94.94 |
| MolProbity score | 1.31 | 2.73 |

[a]Values in parentheses are for highest resolution shell.

the second greatest buried surface area of TACC3 residues at site 1 (97 Å$^2$), and a L532A mutant peptide had 10-fold reduced affinity ($K_d$ = 251 ± 14 μM). In contrast, an R526A mutant peptide bound Aurora-A with an only modestly reduced affinity (49 ± 2 μM), consistent with its contribution to the interaction being through the main chain (Fig EV2B). Mutations within the pocket on Aurora-A which accommodates the F525 side chain (E134A, R137A, L149A, R151A and I158A mutations) all reduced the affinity for the WT TACC3 522–536 peptide. The strongest effect was observed for R151A, which reduced affinity by 40-fold ($K_d$ = 771 ± 61 μM; Fig 2D).

We have previously shown that TACC3 stimulates the activity of Aurora-A, and we therefore investigated the contribution of the interface to this process (Burgess *et al*, 2015). In a $^{32}$P-ATP kinase assay,

WT TACC3[N-ACID] stimulated Aurora-A activity by eightfold, while F525A or L532A mutants stimulated activity by only twofold–threefold (Fig EV2C). The Aurora-A R151A mutation that disrupted the binding interaction to the greatest extent resulted in reduced stimulation by TACC3. These key interaction residues are conserved between TACC3 and Aurora-A orthologs (Fig EV2D). To characterize how TACC3 activates Aurora-A, a kinase assay was performed using unphosphorylated Aurora-A kinase domain and Aurora-A D274N (kinase-dead) as a substrate (Fig 2E). WT TACC3[N-ACID] stimulated both Aurora-A autophosphorylation and substrate phosphorylation on T288. These phosphorylation events were reduced in the presence of TACC3[N-ACID] F525A indicating TACC3 is an allosteric activator of the kinase and enhances substrate phosphorylation.

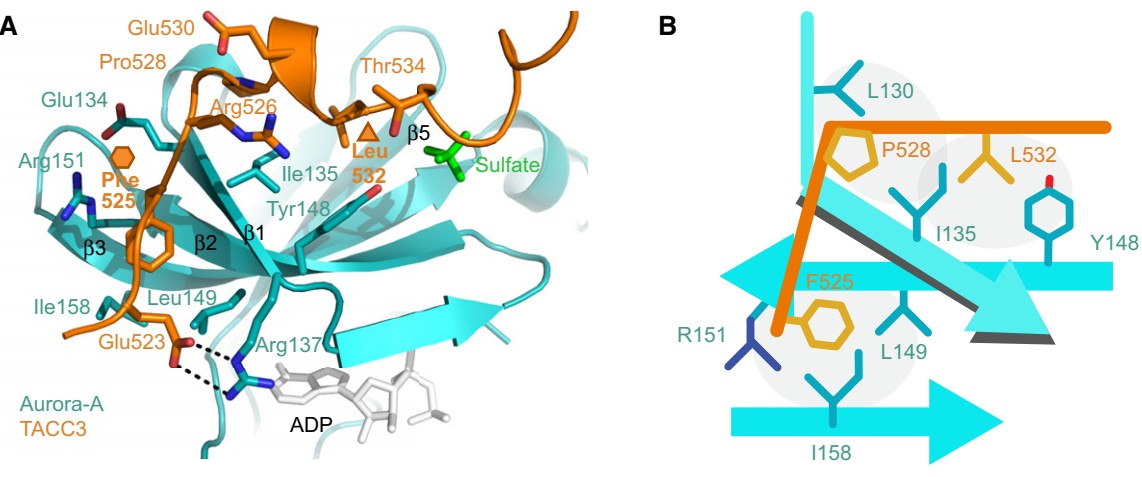

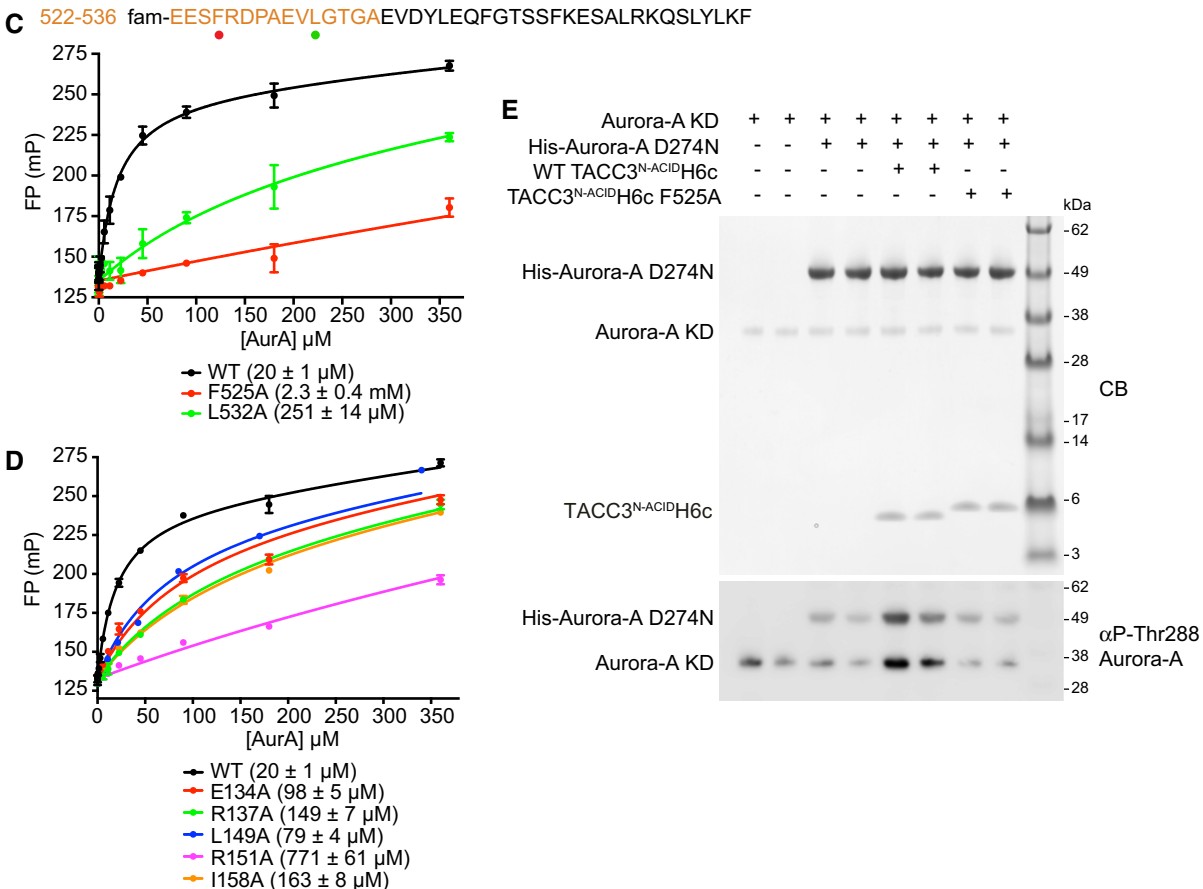

**Figure 2.  TACC3 523–534 forms a docking interaction on the N-lobe of Aurora-A.**

A   Magnified view of the site 1 interaction.

B   Schematic showing the key features of the TACC3/Aurora-A interface.

C   FP assays of binding between site 1 of TACC3 and Aurora-A. The sequence of TACC3 present in the complex crystal structure is shown (top) with the residues of the site 1 FP peptide coloured orange and mutated residues identified by coloured circles corresponding to the curves shown in the FP assay (below). Data represent mean of three experiments ± SD. The binding affinity between WT Aurora-A and FAM-TACC3 522–536 was 20 ± 1 μM. L532A and F525A mutations resulted in reductions in affinity to 251 ± 14 μM and 2.3 ± 0.4 mM, respectively.

D   FP assays between FAM-labelled TACC3 site 1 peptide and Aurora-A mutants (as labelled). Data represent mean of three experiments ± SD. The binding affinity between WT Aurora-A and FAM-TACC3 522–536 was 20 ± 1 μM. The affinities between FAM-TACC3 522–536 and the Aurora-A point mutants were all reduced compared to the WT. The E134A mutant had an affinity of 98 ± 5 μM; R137A, 149 ± 7 μM; L149A, 79 ± 4 μM; R151A, 771 ± 61 μM; and I158A 163 ± 8 μM for TACC3.

E   Kinase assay to monitor autophosphorylation of unphosphorylated Aurora-A KD and substrate phosphorylation of kinase-dead Aurora-A D274N alone and in the presence of TACC3. Reactions were analysed by SDS–PAGE (above) and Western blotting with an αphospho-T288 Aurora-A antibody (below).

Based upon NMR and biochemical data, we conclude that the interaction of TACC3 in solution is that observed at site 1 in the crystal structure and that this site contributes to the activation of Aurora-A by TACC3.

### A model for allosteric activation of Aurora-A by TACC3

In the complex with TACC3$^{N-ACID}$, Aurora-A$^{M3KD}$ is in an inactive conformation in which the αC-helix is distorted and K162 and E181 are separated and unable to form a salt bridge (Fig 3A). Furthermore, the path of the activation loop diverges from that of an active kinase and is disordered between aa281 and aa289. This is consistent with the inactive biochemical state of the Aurora-A$^{M3KD}$ in the complex, which was not phosphorylated on T288. In contrast, the crystal structure of T288-phosphorylated Aurora-A$^{KD}$ in complex with TPX2$^{1-43}$ reveals an active conformation with a salt bridge between L162 and E181 and an ordered activation loop, similar to that of the canonical kinase PKA (Fig 3B and C; Knighton *et al*, 1991; Bayliss *et al*, 2003). The DFG-motif of Aurora-A$^{M3KD}$ in complex with TACC3$^{N-ACID}$ adopts a twisted conformation that directs the activation loop up towards the N-lobe. However, the side chain of Phe275 is located in the expected position for an active, DFG-in conformation, as in the Aurora-A/TPX2 structure (Fig 3D). This contrasts with previous structures of inactive Aurora-A with a distorted αC-helix, in which the side chain of Phe275 occupies a pocket in the N-lobe (DFG-up), forming interactions that may stabilize the distorted conformation of the αC-helix (Fig 3E). Although Aurora-A can adopt an active conformation when unphosphorylated (e.g. PDB code 1MQ4), this reflects the dynamic nature of the kinase, which can be trapped in inactive or active states depending on crystallization conditions. We are therefore cautious about drawing conclusions regarding the mechanism of activation based on the conformation of the kinase that might be influenced by crystal packing interactions (such as site 2 and site 3). Nevertheless, the conformation of Aurora-A in the presence of TACC3 appears to be an intermediate step towards activation, and we speculate that this is how TACC3 allosterically activates Aurora-A. Notably, the opening in which the activation loop sits is widened by movement of the Gly-rich loop, formed from the β1 and β2 strands that interact with TACC3 (Fig 3D). This could lower the energetic barrier to movement of the Phe side chain of Aurora-A, opening space for the αC-helix to reposition and the Lys-Glu salt bridge to form as a precursor to autophosphorylation (Fig 3F). This contrasts with the mechanism by which TPX2 activates Aurora-A, through direct interactions with the αC-helix (Zorba *et al*, 2014; Bayliss *et al*, 2017).

The TACC3 binding site on Aurora-A is distinct from the TPX2 binding site and has not previously been identified as a site of protein–protein interactions in this kinase. We therefore examined the corresponding site in the structures of other kinases (Fig EV3A–E). The β-sheet which in Aurora-A forms the TACC3 binding site is a conserved structural feature of the kinase N-lobe. However, the residues that line the binding site for TACC3 are not conserved: G136 of Aurora-A is replaced by a bulky residue in most kinases, blocking the pocket that is recognized by F525 of TACC3 (Fig EV3A); many kinases have a "N-lobe cap" sequence that extends from the N-terminus of the kinase domain and blocks the

"N-cap" pocket that in Aurora-A remains open for recognition by L532 of TACC3 (Thompson *et al*, 2009). The N-cap pocket is also blocked in the AGC family of kinases (Fig 3C; Knighton *et al*, 1991). TACC3 mimics the interaction of part of the C-terminal extension of PKA with its core kinase domain (N-cap pocket, Fig 3A and C). Similarly, TPX2 binding to Aurora-A resembles the way that other parts of the C-terminal extension (HM pocket, N-cap pocket) and the N-terminal extension (at the W pocket) interact with the core kinase domain of PKA. These observations support the concept of Aurora-A having an incomplete kinase domain that is complemented by regulatory protein binding partners, when compared to PKA, which has an intrinsically active kinase domain because these structural features are encoded into N- and C-terminal extensions of the same polypeptide. The same concept applies to Aurora-B and Aurora-C, which are complemented by the regulatory protein INCENP (Sessa *et al*, 2005). The binding site for TACC3 on Aurora-A is conserved on Aurora-B and Aurora-C, and partly overlaps with the INCENP binding site: the pocket on Aurora-A into which F525 of TACC3 binds is equivalent to that on Aurora-B/C that fits W801 (*Xenopus laevis* numbering; Fig EV3F and G).

Aurora-A in complex with TACC3 has an activation loop conformation that is incompatible with binding of peptide substrates, which explains why the region of TACC3 around S558 is not bound close to the active site. This observation raises the question of whether a single molecule of TACC3 could span both the docking site 1 and the substrate binding site of Aurora-A. To address this point, we generated a model of TACC3 bound to active Aurora-A using FlexPepDock Server (Raveh *et al*, 2010; London *et al*, 2011) and the structure of an active Aurora-A, phosphorylated on Thr288 in complex with TPX2 (Fig 3G). The surface of Aurora-A in the vicinity of the TACC3 site 1 is virtually identical between active and inactive states of the kinase, and the TACC3 docking peptide, aa522–537 adopted a stable conformation, similar to that found in the crystal structure. We modelled the interaction of the substrate region of TACC3 (aa553–561) based on the structure of a substrate-like inhibitor bound to PKA. The TACC3 sequence was readily accommodated into the equivalent binding pocket of Aurora-A. The 16-residue gap between the docking and substrate motifs is more than sufficient to span the distance of 24 Å between their binding sites on Aurora-A, as an extended peptide has a spacing of 3.5 Å per residue.

In summary, Aurora-A recognizes TACC3 through a docking motif to a site of allosteric regulation on the kinase N-lobe. The site has not been reported as a docking site for substrates, but having this site blocked in most kinases means that it is a selective site for binding and regulation of kinases like Aurora-A. We next investigated the functional consequences in cells of disrupting this interaction.

### Abrogation of the interaction that docks TACC3 to Aurora-A results in delayed anaphase onset and cytokinesis

Previous studies have revealed that the localization of Aurora-A to spindle microtubules depends on TPX2 (Bird & Hyman, 2008) and that the phosphorylation by Aurora-A of TACC3 S558 drives its interaction with CHC to form a complex that localizes to K-fibres (Kinoshita *et al*, 2005; Peset *et al*, 2005; Fu *et al*, 2010; Lin *et al*,

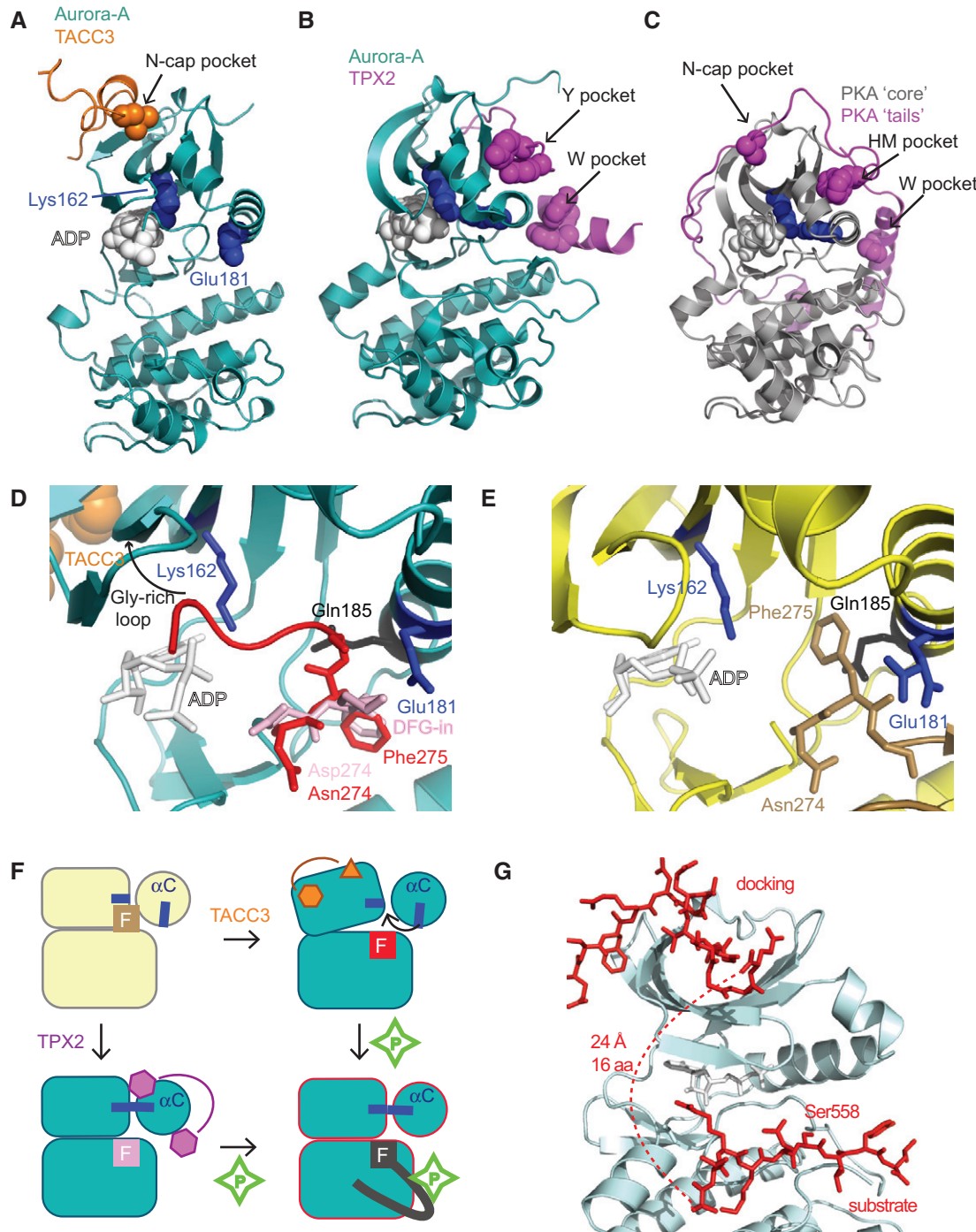

**Figure 3.  Structural analysis of the Aurora-A/TACC3 complex.**

A   Aurora-A^M3KD/TACC3^N-ACID complex, coloured as in Fig 2A. K162 and E181 (blue spheres) do not form a salt bridge, consistent with an inactive state of the kinase.
B   Aurora-A^KD/TPX2^1–43 complex. K162 and E181 (blue spheres) form a salt bridge, consistent with an active state of the kinase (PDB 1OL5).
C   Structure of PKA (PDB 1JBP) with the core kinase domain (grey) and N- and C-terminal extensions (magenta). K72 and E91 (blue spheres) form a salt bridge, consistent with an active state of the kinase.
D   Magnified view of the Aurora-A^M3KD/TACC3^N-ACID complex, centred on the activation loop (red), on which the activation loop from the Aurora-A/TPX2^1–43 structure (pink, PDB 1OL5) is superposed.
E   Magnified view of the Aurora-A^M3/vNAR-D01 complex, centred on the activation loop (brown), which adopts a DFG-up conformation.
F   Schematic illustration of the mechanisms through which TPX2 and TACC3 activate Aurora-A.
G   Computational model of Aurora-A in its active conformation (based on PDB 1OL5) in complex with docking and substrate regions of TACC3.

Data information: Black arrows in (A–C) indicate hydrophobic contacts that are in structurally conserved locations.

2010; Hood *et al*, 2013). All three proteins are localized to spindle microtubules, but it is not known whether the complexes of Aurora-A with TPX2 and TACC3 are co-localized. We investigated the specific localization and timings of the Aurora-A/TACC3 and Aurora-A/TPX2 interactions using proximity ligation assays (PLA) assays. HeLa cells were synchronized and imaged at time points through G2 and M phases (Figs 4A and EV4A and B). In G2, there were few Aurora-A/TACC3 foci across the cytoplasm, but by prophase they accumulated near the condensed chromatin. During prometaphase, the foci appeared in prominent clusters that resembled the spindle poles and by metaphase they had spread along the length of spindle microtubules in a punctate pattern. Aurora-A/TACC3 complexes remained associated with spindle poles throughout anaphase and telophase, although a subset of complexes relocated to the central spindle/midbody and the overall number of PLA foci was reduced. Aurora-A/TPX2 and Aurora-A/TACC3 PLA foci showed a similar localization until late anaphase when Aurora-A/TPX2 relocated entirely to the central spindle/midbody while Aurora-A/TACC3 complexes were retained at the spindle poles, consistent with the localization of TPX2 and TACC3. The distribution of Aurora-A does not perfectly coincide with either of these binding partners (Fig EV4C–E)—it is more concentrated at spindle poles until cytokinesis, when some relocates to the central spindle, as previously reported (Reboutier *et al*, 2013).

Next, we investigated the contribution of site 1 to the localization of TACC3/Aurora-A complexes in HeLa cells. The F525A point mutation was introduced into *TACC3* in HeLa cells by CRISPR/Cas9-mediated genome editing. In addition to a cell clone with the desired F525A mutation (F525A-HeLa), we generated single clones of control (WT) and TACC3 protein null (ΔTACC-HeLa) cells (Appendix Fig S1). Sequence analysis of the *TACC3* transcripts present in the F525A-HeLa clone revealed three transcript species in total. The majority (~80%) was full-length and carried the F525A point mutation. The smaller group was split between transcripts encoding a protein product truncated at exon 5, and a product containing the E519D mutation along with an in-frame deletion of site 1 (aa520–529). The former is unlikely to be produced in cells, since no truncated proteins were detected on Western blots probed with an antibody against an exon 3-encoded portion of TACC3 (Appendix Fig S1C). However, the site 1-deleted form may be expressed alongside the TACC3-F525A product in the F525A-HeLa cells, but for simplicity we refer to the TACC3 product in this cell line as TACC3-F525A. When compared to wild-type TACC3, TACC3-F525A seemed markedly reduced on mitotic spindles during all stages of mitosis (Figs 4B and EV4C). Because levels of wild-type TACC3 and TACC3-F525A were comparable on Western blots, the F525A mutation impairs spindle targeting and/or retention rather than protein expression or stability (Appendix Fig S1C). Given the crucial role of F525 within site 1 (i.e. the Aurora-A docking site) of TACC3 *in vitro*, we next assayed its contribution to Aurora-A/TACC3 complex formation in mitotic cells. In line with the overall reduction in TACC3 observed on the spindle, PLA-based quantification revealed 50% fewer complexes in both early and late mitotic F525A-HeLa cells (Fig 4C and D).

We next performed time-lapse imaging of mitosis in WT, F525A-HeLa and TACC3 null ΔTACC-HeLa cells (Figs 4E and EV4F–H). Similar to ΔTACC-HeLa, F525A-HeLa progressed more slowly from nuclear envelope breakdown (NEBD) to anaphase onset than WT

(Fig EV4H). Moreover, the time between anaphase and completion of chromosome decondensation was also longer in ΔTACC and F525A-HeLa cells (time in minutes: ΔTACC-HeLa: 48 ± 9, F525A-HeLa: 49 ± 11, WT: 37 ± 6; Fig 4E). The delay occurred mostly between telophase and chromosome decondensation (Fig EV4F). These findings indicate a key role for site 1, and the F525 residue in particular, during both early and late mitosis (Burgess *et al*, 2015).

In summary, our results confirm that the site 1 interface observed in the crystal structure is critical for Aurora-A/TACC3 interaction and also for proper TACC3 localization in mitotic cells.

## Crystal structure of a chimeric fusion of CHC with pTACC3

The complex between CHC and TACC3 cross-links microtubules contributing to the formation of K-fibres and the fidelity of mitosis. Previous work mapped the CHC-binding region of TACC3 to aa549–570, which includes the S558 site that must be phosphorylated by Aurora-A to drive the interaction, but not the Aurora-A docking motif centred on F525 (Fig 5A; Hood *et al*, 2013; Burgess *et al*, 2015). We set out to determine the structural basis of the interaction between TACC3 and CHC. To help plan our strategy, we first measured the binding affinity between a CHC propeller-ankle fragment and FAM-labelled TACC3 peptides using FP.

We initially trialled different length CHC fragments and TACC3 peptides phosphorylated on S558 (Figs 5B and EV5A and B). We then explored the contribution of TACC3 phosphorylation to binding affinity (Figs 5C and EV5C). Non-phosphorylated TACC3$^{CID}$ (aa549–570) bound weakly to CHC aa1–574 and, while incorporation of a phospho-mimic S558E mutation did not increase affinity, phosphorylation of S558 enhanced affinity fourfold. Based on these observations, we generated a fusion protein (ClACC) incorporating CHC aa1–574 and TACC3$^{CID}$ separated by four Thr-Gly-Ser linkers (TGS$_4$) with full incorporation of phosphoserine at TACC3 residue 558 (ClACCp) using a genetic encoding system (Appendix Fig S2A–D; Rogerson *et al*, 2015). Crystals of ClACCp protein diffracted to 3.09 Å, and the structure was solved by MR using a model of CHC 1–494 (PDB 1BPO; ter Haar *et al*, 1998; Table 1).

The ClACCp structure had four chains of CHC aa1–574 and four chains of TACC3 aa550–567 in the asymmetric unit (ASU). The TGS$_4$ linker was not resolved (Appendix Fig S2E–J). The residues of CHC not present in the search model form an additional α-helical repeat comprising six helices (α11–α16, Fig 5D). Superposition of the four complexes present in the ASU by alignment of the CHC β-propeller reveals differences between them in the positions of helices in the linker and ankle regions (Appendix Fig S2F) due to the flexibility of these regions. However, when we aligned the ankle region of the four molecules in the ASU, they were found to superimpose with C$_\alpha$ RMSD ranging from 0.7 Å to 1.2 Å, and the mode of interaction of the ankle region with pTACC3 was equivalent in each (Appendix Fig S2G).

## CHC recognizes hydrophobic residues of TACC3 that are displayed on an α-helix

The core of the interface encompasses TACC3 aa559–565, and the α10 and α12 helices of CHC (Fig 5D–F). Surprisingly, pSer558 is not at the core of the interface: the nearest functional group is the

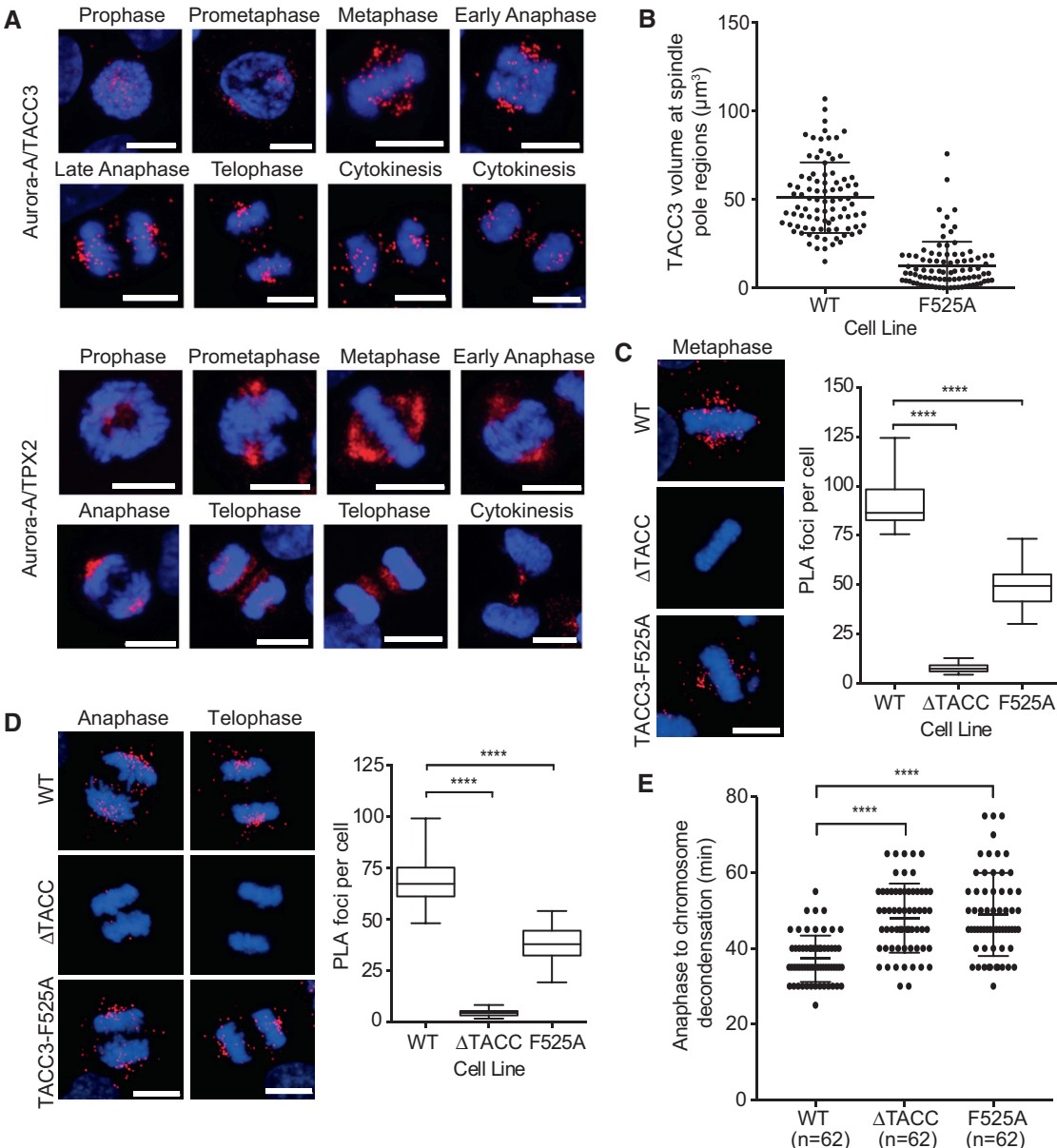

**Figure 4. Defects in spindle localization and late mitotic progression in CRISPR/Cas9-engineered HeLa cells with mutations in TACC3.**

A   PLA showing Aurora-A/TACC3 (top panel) and Aurora-A/TPX2 (bottom panel) complexes during mitosis in HeLa cells. Cell cycle stages are indicated above cell images.

B   Dot plot indicates volume of TACC3 protein at spindle pole regions of WT and F525A HeLa cells in telophase. Volumes were measured for 90 spindle pole regions per cell line (45 telophase cells). Representative images are shown in Fig EV4C.

C, D   PLA between Aurora-A and TACC3 in WT, ΔTACC and F525A HeLa cells. Representative metaphase (C), and anaphase and telophase cells (D) are shown in panels on the left. Box and whisker plots on the right represent the number of PLA dots counted per cell in either metaphase or anaphase/telophase cells, respectively. 30 cells were analysed for each cell cycle stage for each cell line.

E   Duration of anaphase to chromosome decondensation, measured by time-lapse microscopy in WT, ΔTACC and F525A HeLa cells. Number of cells analysed is indicated on graph (*n*).

Data information: Cell images include DNA stained with DAPI (blue). Scale bars, 10 μm. In the dot plots (B, E), horizontal lines represent the mean and error bars correspond to standard deviation. *P*-values were obtained from Mann–Whitney: ****P* < 0.0001. For the box and whisker plots (C, D), the middle horizontal line marks the median and the box represents the 25th and 75th percentile. The whiskers extend from the minimum to the maximum values. *P*-values were obtained from Student's *t*-test, unpaired, two-tailed: ****P* < 0.0001.

amine on the side chain of CHC K507, which is ~4 Å away but the weak electron density of this residue suggests mobility, not a stable interaction. The core of the interface is hydrophobic, and the side chains of TACC3 residues L559, Y560, F563, and P565 pack against the side chains of CHC residues L476, L480, R481, L503, Y504, and K507. The side chain of R481 stacks on the aromatic ring of TACC3

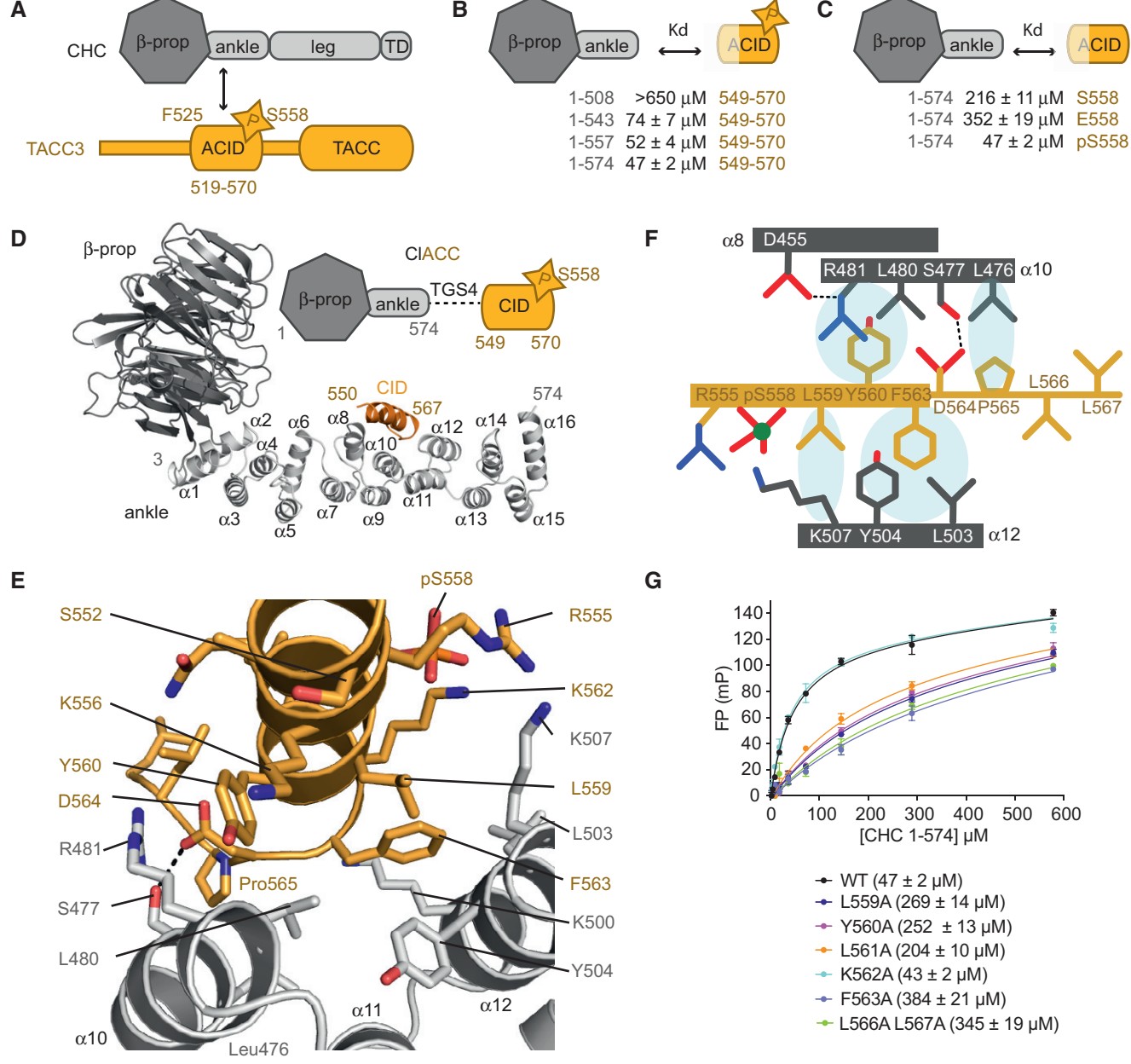

**Figure 5.  Crystal structure of ClACCp.**

A  Schematic illustration of CHC (grey) and TACC3 (orange) domain structures.

B  Summary of the binding data between CHC proteins and TACC3[CID] peptides. See also panel (G) and Fig EV5A and B.

C  FP assays to measure the binding affinity of CHC 1–574 to FAM-labelled TACC3[CID] phosphorylated (pS558), unphosphorylated (S558) and phosphomimetic (S558E) forms. See also Fig EV5C.

D  Overview of the crystal structure of ClACCp. Note that the linker between CHC aa574 and TACC3 aa550 is disordered.

E  Magnified view of the CHC/TACC3 interaction. The side chains of key interacting residues are shown as sticks.

F  Schematic illustration of the CHC/TACC3 interface.

G  FP binding curves of CHC 1–574 with FAM-phospho-TACC3[CID] bearing alanine point mutations. Affinities are in parentheses. Data represent mean of three experiments ± SD.

Y560 and is held in position by the side chain of D455 from α8. In addition, there is a H-bond between the side chains of TACC3 D564 and CHC S477. Substitutions to alanine of residues L559, Y560, and F563, from the core of the interface with CHC, in FAM-labelled

pTACC3[CID] peptides each reduced the affinity for CHC 1–574 in FP assays by at least fourfold compared to WT (Fig 5G). Thus, these three hydrophobic side chains each contribute to the interaction, consistent with the crystal structure. However, mutation of L566/

**Figure 6.  Mutations in the TACC3/CHC interface reduce spindle localization of the complex.**

A   Cartoon representation of the TACC3/CHC interface (orange and grey, respectively). Key CHC residues are shown as spheres.

B   FP binding curves of CHC 1–574 mutants with FAM-phospho-TACC3$^{CID}$. Affinities are in parentheses. Data represents mean of three experiments $\pm$ SD.

C   Representative confocal images of HeLa cells at metaphase expressing the indicated GFP-TACC3 construct on a background of TACC3 depletion and stained for tubulin and CHC. Below, scatter plot to show mitotic spindle enrichment of TACC3 constructs (red) and endogenous CHC (blue); GFP controls are pale red. ANOVA with Tukey's *post hoc* test, only comparison of TACC3 constructs with WT is shown. *$P < 0.05$, ***$P < 0.001$.

D   Representative confocal images of HeLa cells at metaphase expressing the indicated GFP-CHC construct on a background of CHC depletion and stained for tubulin and TACC3. Below, scatter plot to show mitotic spindle enrichment of CHC constructs (blue) and endogenous TACC3 (red); GFP controls are pale blue. ANOVA with Tukey's *post hoc* test, only comparison of CHC constructs with WT is shown. **$P < 0.01$, ***$P < 0.001$.

Data information: Each spot represents the average measurement per cell. Bars show mean values $\pm$ standard deviation. Scale bar, 10 μm.

L567 and L561 to alanine also reduced the binding affinity of the interaction, while these residues do not make substantial direct contact with CHC (Fig 5G). We rationalized this with the observation that the side chains of L566 and L561 pack together to stabilize the C-terminus of the helix, which is terminated by P565, and thus may help to position residues such as F563 and P565 on the opposite side of the α-helix that do make substantial contacts with CHC. However, a wider range of altered CHC-binding properties in the TACC3 variant peptides should perhaps be expected and so further functional analysis of the interaction was required to increase confidence in the model of the TACC3/CHC interface.

Next, we examined whether the CHC-TACC3 interaction could be disrupted by mutations on the surface of CHC, based on analysis of the crystal structure (Fig 6A). For example, R481 interacts with both TACC3 Y560 and TACC3 D564 and was mutated to glutamic acid to disrupt both interactions. We explored the opportunities for disruption of specific contacts with the side chain of TACC3 F563: L480 nestles against the Cβ atom of TACC3 F563 and was mutated to aspartic acid alone or in combination with a L476D mutation, which disrupts the hydrophobic surface contacting P565; the double mutation L503A/Y504A removes the pocket into which the phenyl group of F563 binds; the subtle F492Y mutation introduces an –OH group that is predicted to clash with the Cβ atom of TACC3 F563. The mutant CHC proteins were expressed and purified as the WT protein, and binding affinities to the FAM-labelled pTACC3$^{CID}$ were measured using the FP assay (Fig 6B). The mutation F492Y caused > 6-fold increase in $K_d$. The four other mutant CHC proteins were measured to have $K_d$s for pTACC3$^{CID}$ of over 300 μM. We conclude that interaction between CHC and TACC3 in solution is consistent with that observed in the crystal structure.

We next probed whether the TACC3-CHC interface observed in the crystal structure was relevant to the function of the interaction in HeLa cells (Fig 6C and D). We previously showed that the interaction between the two proteins is essential for their recruitment to the mitotic spindle, so here we asked whether the spindle localization was impaired in TACC3 or CHC proteins harbouring mutations in the observed interface.

First, we re-expressed a number of GFP-TACC3 constructs in HeLa cells depleted of endogenous TACC3 and imaged cells fixed during metaphase (Fig 6C). In the control cells, GFP alone exhibited a diffuse, cytoplasmic localization, whereas WT TACC3 was tightly localized to the spindle. A control mutant, in which residues in the vicinity of S558 that do not contact CHC were mutated (L561A, K562A) behaved like WT. In contrast, other mutants (L566A, L567A; F563A; L559A; Y560A) that disrupt the interaction *in vitro* were not localized to the spindle. Furthermore, these mutants were

unable to rescue the localization of endogenous CHC, which has a clear enrichment to the spindle in WT cells, but a more diffuse localization when TACC3 is depleted.

We then re-expressed the designated GFP-CHC constructs in HeLa cells depleted of endogenous CHC. In line with previous experiments, WT GFP-CHC, but not GFP alone showed enrichment at the spindle, although there is a background of cytoplasmic CHC (Fig 6D). All CHC mutants tested were impaired in spindle localization: a control in which the TACC3 interaction site is deleted (Δ457–507); mutations in the TACC3 binding site (L476D, L480D; R481E; L503A, Y504A) and mutation of a residue that holds R481 in place (D455K). However, these data should be interpreted cautiously because rescue of CHC knockdown with exogenous WT-CHC was unable to fully restore TACC3 localization to the spindle. Either the exogenously expressed CHC is impaired in binding TACC3 or the levels of exogenous CHC are too low, and so we cannot draw definitive conclusions from the rescue experiments with CHC mutants. Moreover, the results suggest that the localization of CHC to the spindle has a component that is independent of the interaction we have characterized, most likely the N-terminus of clathrin/TACC domain of TACC3, and other components of the complex such as GTSE1 (Hubner *et al*, 2010; Hood *et al*, 2013).

### Formation of a cryptic α-helix, stabilized by phosphorylation of S558, contributes to CHC binding

We noticed that, in the structure of the pClACC fusion, residues 550–563 of phosphorylated TACC3 form an α-helix (α1P), whereas the corresponding region of unphosphorylated TACC3 bound to Aurora-A forms an α-helix at 546–555 (α1), but residues 556–563 adopt an extended conformation (Fig 7A and B). α1P is an extended form of the shorter α-helix α1 that organizes the hydrophobic residues, such as Y560 and F563, that are critical for the interaction with CHC (Fig 7C).

We used NMR spectroscopy to determine the extent to which these α-helices are present in TACC3 in the absence of binding partners in solution. Chemical shift index (CSI) plots were created for phosphorylated and unphosphorylated TACC3$^{CID}$ following standard methods (Fig 7D and E; Wishart & Sykes, 1994). In addition to calculating α-proton chemical shifts we also measured secondary chemical shifts for α-carbons. After the type of amino acid, the main contribution to chemical shifts of α-carbons is the ψ angle and thus the backbone conformation. Proximity of charges or aromatic side chains which can strongly affect proton chemical shifts has virtually no effect on the chemical shifts of α-carbons which makes them very robust probes of backbone conformation. CSI analysis of

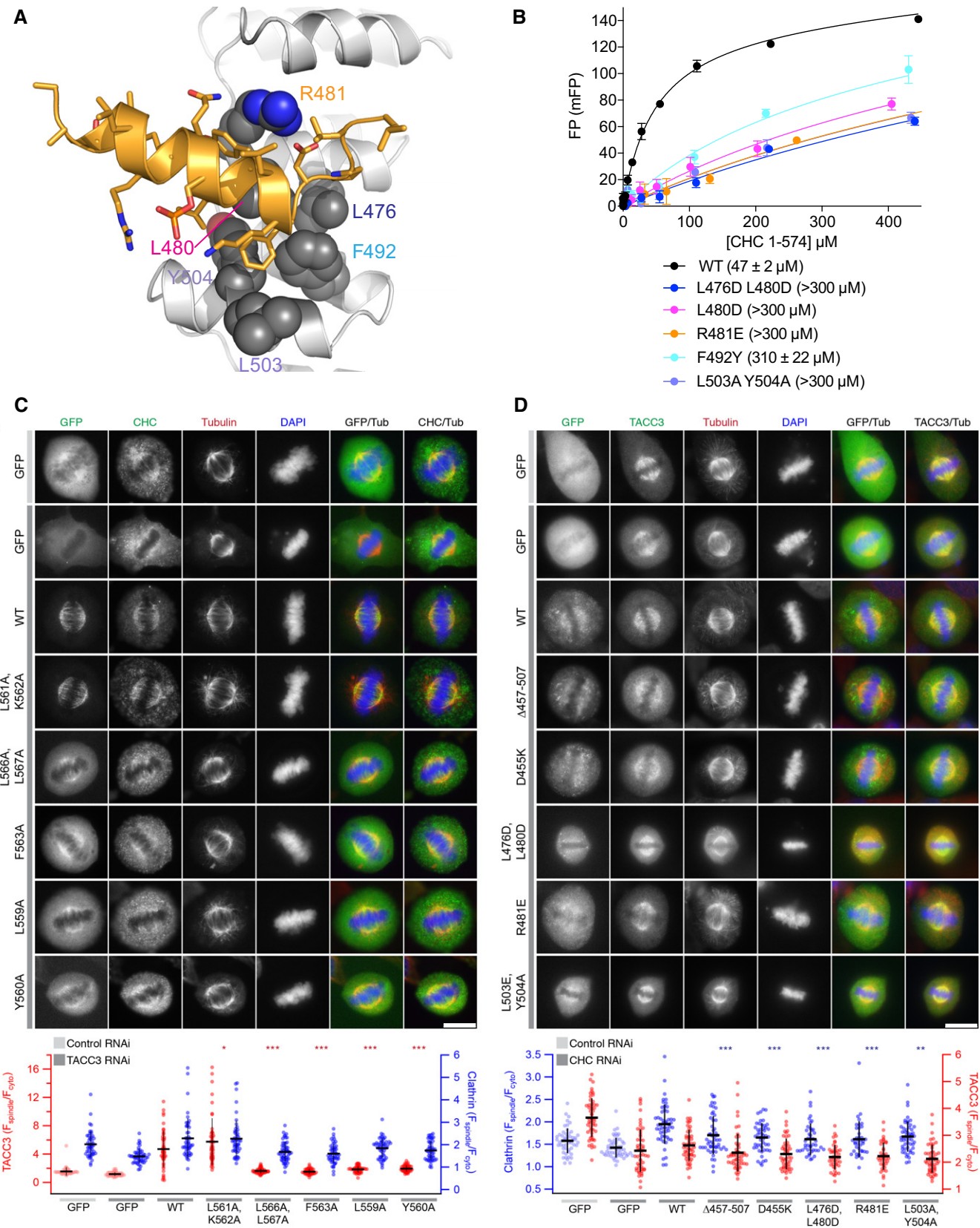

**Figure 6.**

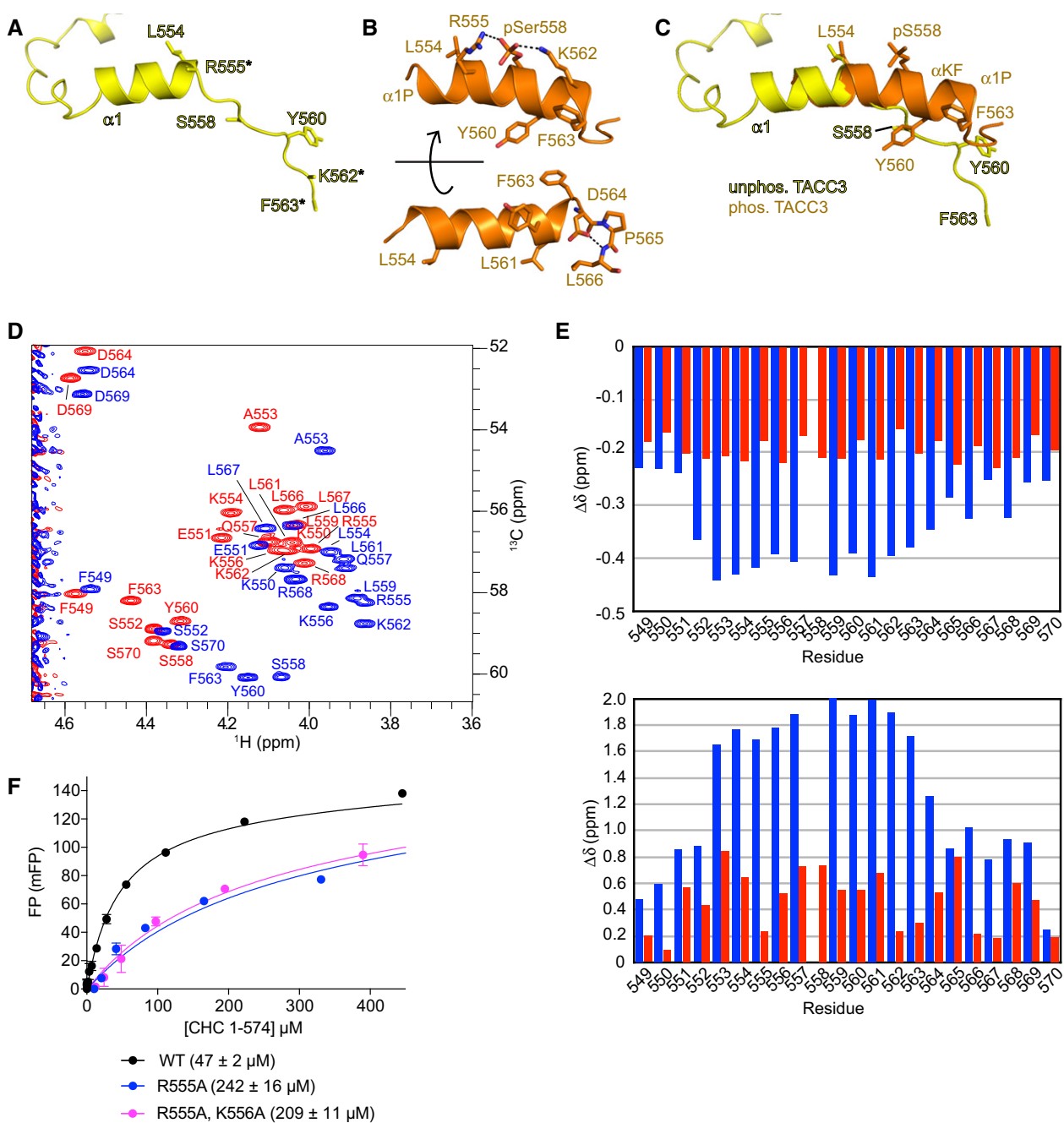

**Figure 7.  Phosphorylation of S558 promotes formation of an α-helix that binds CHC.**

A   Structure of unphosphorylated TACC3[N-ACID] extracted from site 2 of the complex with Aurora-A. Disordered side chains are marked with asterisks.
B   Two views of phosphorylated TACC3[CID] extracted from the complex with CHC. Polar interactions are marked with dashed lines.
C   Superposition of the structures from panels (A) and (B) shows the transition in residues 558–563 from extended conformation to α-helix.
D   Superposition of the [13]C-HSQC spectra of unphosphorylated (red) and phosphorylated S558 (blue) TACC3 549–570 peptides. Shown is the region of the Cα-Hα correlations from which the data for the secondary chemical shifts calculations were derived.
E   Secondary chemical shifts for unphosphorylated (red) and phosphorylated S558 (blue) TACC3 549–570 peptides. Data for αH are shown on the top and αC on the bottom.
F   FP binding curves of CHC 1–574 with WT and mutant FAM-phospho-TACC3[CID]. Affinities are in parentheses. Data represent mean of three experiments ± SD.

synthetic TACC3[CID] peptides indicated a tendency towards helix formation in the unphosphorylated peptide, but a much higher propensity in the phosphorylated peptide, especially within residues 559–564 (Fig 7E). We concluded that phosphorylation of S558 increases the propensity for α-helix formation in TACC3, and we examined the structure to identify the mechanism.

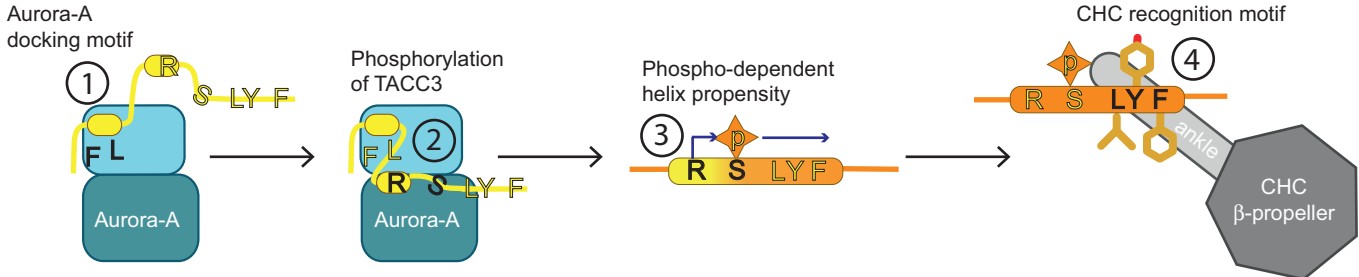

**Figure 8. Schematic model of the interactions and conformational changes in TACC3 during its recruitment to the mitotic spindle.**

(1) TACC3 docks to Aurora-A through a motif centred on F525 and L532. (2) Activated Aurora-A phosphorylates TACC3 on S558. (3) Phosphorylated TACC3 has increased helical propensity. (4) The helical conformation of TACC3 docks to CHC via hydrophobic residues L559, Y560 and F563.

S558 is positioned in-between basic residues, R555 and K562, which are on the same face of the helix and are therefore positioned to form salt bridges with the phosphorylated side chain that could act as a "staple" to stabilize the α-helix (Fig 7B). In addition, the side chain of K556 forms another "staple" through interaction with Ser552. However, the positions of these basic side chains are not clearly resolved in the electron density maps, which are based on diffraction data of modest resolution and high B-factor (Appendix Figs S2E and S3). We probed the contribution of these interactions to the CHC/TACC3$^{CID}$ interaction using the FP assay (Fig 7F). The R555A mutation alone or in combination with K556A reduced the affinity by a factor 4–5, similar to that of the unphosphorylated TACC3$^{CID}$ peptide (216 ± 11 μM; Fig EV5C), whereas the effect of the K562A mutation was negligible (Fig 5G). R555 does not interact directly with CHC, and so we conclude that it contributes to CHC binding through an intramolecular "staple" interaction that stabilizes the extended helix α1P. Taken together, these data indicate that phosphorylation stabilizes a helical conformation of TACC3, which places a series of critical hydrophobic side chains into the correct positions to interact with the complementary hydrophobic surface of CHC.

Structural studies revealed that the ACID region of TACC3 has three separate motifs that mediate molecular recognition events at each step of the pathway (Fig 8). The first motif docks TACC3 to the N-lobe of Aurora-A. The second motif, which is the site of protein phosphorylation, is recognized in its extended conformation by the active site of Aurora-A and then undergoes a conformational transition to form an α-helix. The third motif is recognized by CHC, preferentially in the context of an α-helix, formation of which depends on the previous two steps.

## Discussion

### A late mitotic role for the Aurora-A/TACC3 interaction

Aurora-A contributes to the function of the spindle throughout mitosis, regulating the generation of a robust, bipolar spindle during early mitosis, and contributing to spindle disassembly in late mitosis (Lioutas & Vernos, 2013; Reboutier *et al*, 2013). These temporally distinct functions were found initially by acutely inhibiting Aurora-A using antibody injection of cells at defined time points, resulting in delays and microtubule defects during anaphase (Marumoto *et al*, 2003). Conditional knockout of *AURKA* in chicken DT40 cells

showed that, surprisingly, Aurora-A is dispensable for bipolar spindle assembly, although this process is much slower in its absence, and that the major phenotype results from defects and delays in anaphase (Hegarat *et al*, 2011). More recently, taking advantage of chemical genetic approaches, TACC3 and p150glued were identified as Aurora-A substrates that contribute to microtubule stability and central spindle assembly (Lioutas & Vernos, 2013; Reboutier *et al*, 2013). Here, we show that abrogation of the docking interaction between Aurora-A and TACC3 generates a delay after metaphase. A role for Aurora-A and TACC3 in central spindle assembly and elongation has been described previously (Marumoto *et al*, 2003; Lioutas & Vernos, 2013). Although we detect some delay in anaphase to telophase progression in F525A-HeLa cells, most of this occurs between telophase and cytokinesis in both the TACC3 null and F525A-HeLa cells. The reduced spindle localization of the F525A mutant in HeLa cells is similar to what we previously observed with the F525A equivalent mutation in chicken DT40 cells, but there is a critical difference in the mitotic timings between these cellular systems. The F525A equivalent mutation in DT40 cells shortened the duration of NEBD to anaphase onset by ~5 min, whereas F525A-HeLa cells exhibited a delay of ~35 min compared to single-cloned controls, suggesting increased reliance of spindle assembly in HeLa cells on TACC3-containing inter-microtubule bridges. Indeed, these bridges are likely to play a more central role in stabilizing human K-fibres that contain 20–40 microtubules, than DT40 K-fibres, which comprise only ~4 microtubules (Ribeiro *et al*, 2009; Nixon *et al*, 2015). The localization of TACC3 to spindle microtubules depends on Aurora-A kinase activity, which is mainly driven by its interaction with TPX2, and is antagonized by PP6 (Eyers *et al*, 2003; Tsai *et al*, 2003; Manfredi *et al*, 2007; Zeng *et al*, 2010). In turn, TACC3 localization is dependent on phosphorylation (Giet *et al*, 2002; Kinoshita *et al*, 2005; Peset *et al*, 2005). Our results therefore show that the docking interaction between Aurora-A and TACC3 also makes a major contribution to TACC3 spindle localization. This interaction increases the efficiency of TACC3 phosphorylation by both enhancing Aurora-A catalytic activity and tethering the kinase close to the S558 phosphorylation site. These effects can contribute throughout mitosis; however, the ability of TACC3 to both dock to Aurora-A and activate the kinase locally appears to be most critical in late mitosis when it is physically separated from the pool of TPX2-activated Aurora-A. We have begun to probe this model; however, it is challenging to study the dynamic and localized activity of Aurora-A in cells because activation loop phosphorylation on T288, which can be detected by antibody or

biosensor, is not an absolute marker of activity and unphosphorylated kinase can also be highly active (Dodson & Bayliss, 2012; Bertolin *et al*, 2016).

### Structural insights into clathrin flexibility and interactions

The structure of CHC is best characterized as coat assemblies formed by arrays of CHC-triskelia (Kirchhausen, 2000; Brodsky, 2012). The CHC triskelion is a three-legged structure, formed by three heavy chains, each associated with a light chain. CHC contains an N-terminal β-propeller domain and an extended region containing helical repeats categorized into linker, ankle, distal leg, knee, and proximal leg, while the C-terminus has a trimerization domain (Appendix Fig S4A; Fotin *et al*, 2004). Cryo-EM studies have described the structure of CHC to 7.9 Å (Fotin *et al*, 2004); however, atomic resolution structures are known only for the β-propeller domain and linker (aa1–494; ter Haar *et al*, 1998) and a part of the proximal leg (aa1210–1516; Brodsky, 2012) and there is a near-atomic resolution crystal structure of the trimerization domain (aa1521–1654; Ybe *et al*, 2013). Here, we extend the knowledge of CHC structure at high resolution to residues 1–574. The conformation of the ankle region of CHC is more compact in the ClACC structure compared to the previous structure of CHC 1–494, which could be a consequence of TACC3 binding between two helices of the repeat (Appendix Fig S2I and J). It is worth noting that this binding site could be exploited by other CHC-binding partners. Indeed, we noticed that the auxilin-binding site within the CHC lattice lies in close proximity to the TACC3-interacting region of the ankle domain (Appendix Fig S4B). If auxilin and TACC3 do share a common binding site on CHC, this could explain an earlier observation in which auxilin depletion resulted in sequestration of TACC3 in CHC-coated vesicles (Borner *et al*, 2012).

### A site for substrate docking and allosteric regulation on the N-lobe of Aurora-A

We have resolved why a region 30aa N-terminal to the phosphorylation site is critical for TACC3 binding to Aurora-A. The region of TACC3 that is phosphorylated has low affinity for Aurora-A, and the binding affinity of this substrate is primarily through a docking motif that also stimulates kinase activity. Substrate docking interactions that allosterically regulate kinase activity through conformational changes have been previously observed in MAP kinases and AGC kinases (Goldsmith *et al*, 2007). Interestingly, three different substrates of Aurora-A (TACC3, TPX2, and N-Myc) have three distinct docking sites that can all be used to stimulate kinase activity. Other kinases engage the N-cap pocket as a site for regulation. For example, the SH2 domain of the tyrosine kinase Abl acts as an allosteric activator through interactions at the N-cap pocket, and I164 of Abl fulfils a structurally equivalent role to that of TACC3 L532 (Fig EV3D; Nagar *et al*, 2006; Filippakopoulos *et al*, 2008). The N-cap pocket is a binding site for small-molecule allosteric activators of AMPK (Xiao *et al*, 2013; Calabrese *et al*, 2014). However, the closest analogy to the Aurora-A/TACC3 interaction is CK2, which comprises a catalytic subunit (CK2α) bound to a dimer of noncatalytic subunits (CK2β), which can be recapitulated using a short, cyclic peptide (Pc) based on CK2β (Niefind *et al*, 2001; Raaf

*et al*, 2013). A crystal structure of CK2α/Pc shows a remarkable similarity with the Aurora-A/TACC3 interface, with F190 of Pc/CK2β taking the role of TACC3 L532 (Fig EV3E). AGC kinases have a C-terminal region extension that wraps around the catalytic domain, and inserts a hydrophobic residue into the N-cap pocket, such as I335 of PKA (Knighton *et al*, 1991). However, the role of these interactions in promoting kinase activity has been enigmatic. Here, we propose that TACC3 activates Aurora-A through an interaction at the N-lobe pocket by destabilizing an inactive conformation of the kinase in which the side chain of Phe275 (from the DFG-motif) blocks the C-helix in a distorted conformation. This mechanism is similar in principle to the activation of tyrosine kinases through conformational changes that facilitate the flip of the DFG-motif (Levinson *et al*, 2006), and we speculate that a similar mechanism may contribute to the activation of other kinases through interactions at their N-lobe pockets.

### A flexible, non-canonical mechanism of phospho-recognition

We have resolved how phosphorylated TACC3 is handed over from Aurora-A to CHC (Fig 8). Aurora-A phosphorylates S558 in the context of an extended peptide conformation, which is necessary for it to be accommodated in the cleft that recognizes substrates. Phosphorylation at S558 triggers a localized disorder to order transition, stabilizing and extending a nascent helix. The structural basis of TACC3 binding to CHC is unusual in that CHC recognizes the structural consequence of phosphorylation. This explains how CHC recognizes phosphorylated TACC3 even though it does not have a canonical pSer binding domain. We propose that the α1P-helix of TACC3 is stabilized through formation of an intramolecular salt bridge with R555 that acts as a staple to hold this region of TACC3 in a helical conformation. Salt bridges formed between glutamate and lysine/arginine residues stabilize long helices in the context of naturally occurring single alpha-helix (SAH) domains, such as those found in myosins 6 and 10, or synthetic model SAH domains (Wolny *et al*, 2017). These give rise to constitutively α-helical proteins, however, in TACC3, the formation of the salt bridge is regulated by Aurora-A kinase. The presence of an arginine three residues N-terminal to the phosphorylation site (P-3) is a conserved feature of the substrates of basophilic kinases such as Aurora-A or PKA and is recognized by an acidic pocket in the C-lobe of these kinases (Knighton *et al*, 1991). Therefore, we expect that the stabilization of helices by phosphorylation could be a recurring mechanism for the regulation of protein–protein interactions by basophilic kinases.

## Materials and Methods

### DNA manipulation

Expression vectors, pET30TEV Aurora-A 122–403 C290A, C393A, pET30TEV Aurora-A 122–403, pET30TEV Aurora-A 1–403 D274N, pETM6T1 TACC3 519–563, and pET44c TACC3 519–563-H6c, were produced in earlier work (Bayliss *et al*, 2003; Burgess *et al*, 2015; Rowan *et al*, 2013).

The expression construct, pETM6T1 TACC3 519–540, was produced by the introduction of a stop codon into the vector, pETM6T1 TACC3 519–563 at residue 541 by site-directed

mutagenesis using PfuTurbo polymerase (Agilent). Point mutations in pET44c TACC3 519–563-H6c and pET30TEV Aurora-A 122–403 C290A, C393A were produced in a similar manner.

For co-expression with pETM6T1 TACC3 519–563, the coding sequence of Aurora-A 122–403 D274N, C290A, C393A was subcloned into pCDF.

Overlapping extension PCR was performed to make the chimeric construct CHC 1–574-TGS$_4$-TACC3 549–570$^{S558TAG}$-TEV site 6xHis-Tag (ClACCp). The construct was subcloned in pNHD (Rogerson *et al*, 2015) using the restriction sites BsrG1 and Xho1 to produce the vector, pNH-ClACCp. The amber codon at S558 was introduced into TACC3 by site-directed mutagenesis for incorporation of synthetic phosphoserine by orthogonal aminoacyl-tRNA synthetase/tRNA$_{CUA}$ pairs via the genetic code expansion method (Rogerson *et al*, 2015).

For binding studies, CHC constructs were cloned in a modified pGEX vector containing a TEV-cleavable GST tag and point mutations introduced by site-directed mutagenesis.

For expression of GFP-CHC or GFP-TACC3 mutants in cells depleted of endogenous proteins, the pBrain system was used (Hood *et al*, 2013). New mutants were generated by site-directed mutagenesis.

## Protein expression and purification

Aurora-A, TACC3, and TPX2 constructs were expressed and purified as described in previous work (Burgess *et al*, 2015).

Double-labelled $^{13}C^{15}N$ TACC3 519–540 was expressed and purified as described for $^{13}C^{15}N$-TACC3 519–563 (Burgess *et al*, 2015). The protein was concentrated using Spectra gel as per the manufacturer's instructions (Spectrum).

To generate diffraction-quality complexes, we used a construct of Aurora-A (Aurora-A$^{M3KD}$) that was mutated in three places: D274N prevents coordination to magnesium, inactivates the kinase, and thus prevents heterogeneous phosphorylation of the proteins, and C290A, C393A stabilizes Aurora-A and improves the diffraction limit of Aurora-A crystals (Burgess & Bayliss, 2015). To produce the Aurora-A/TACC3 complex for crystallization trials, the expression vectors, pCDF Aurora-A 122–403 D274N, C290A, C393A, and pETM6T1 TACC3 519–563 were co-transformed into *Escherichia coli* BL21(DE3)RIL cells and recombinant protein expression and purification carried out as described previously for TACC3 constructs (Burgess *et al*, 2015). As a final purification step, the complex was subject to size-exclusion chromatography using a Superdex 200 16/600 column (GE Healthcare) equilibrated in 20 mM Tris pH 7.0, 200 mM NaCl, 5 mM β-mercaptoethanol, 5 mM MgCl$_2$, and 10% glycerol.

For expression of ClACCp containing unnatural phosphoserine at S558 site on TACC3 549–570, pNH-ClACCp was transformed in *E. coli* BL21 Δ*serB*(DE3) cells containing pKW2-EF-Sep (Rogerson *et al*, 2015). Colonies were grown overnight in terrific broth (TB) containing 30 μg/ml chloramphenicol and 25 μg/ml tetracycline. The overnight cultures were used to inoculate 12 × 1 l TB containing the appropriate antibiotics and 2 mM phosphoserine. The cultures were subsequently incubated at 37°C at 200 rpm. At OD$_{600}$ ~0.7–0.8, 200 μM IPTG and 2 mM phosphoserine were added and the cultures incubated for a further 16 h at 18°C and 200 rpm. The cells were harvested by centrifugation and

resuspended in lysis buffer (50 mM Tris pH 9.0, 200 mM NaCl, 1 mM β-mercaptoethanol) containing 2.5 mM PMSF and lysed by sonication. The cell lysate was clarified by centrifugation and subsequently purified using a HisTrap affinity column (GE Healthcare) equilibrated in lysis buffer and eluted with an imidazole gradient (20–400 mM). The protein was dialysed against 20 mM Tris pH 9.0, 200 mM NaCl, 1 mM EDTA, 5 mM DTT at 4°C in the presence of His-tagged TEV protease to remove the C-terminal His-tag. The cleaved His-tag and His-tagged TEV protease were removed by pass-back over HIS-Select Nickel Affinity Gel (Sigma). The protein was further purified by anion exchange chromatography using 5 ml HiTrap Q HP column (GE Healthcare) equilibrated in 20 mM Tris pH 9.0, 5 mM DTT and eluted with increasing salt concentration (1 M). The complex was polished by size-exclusion chromatography using a Superdex-200 16/600 column (GE Healthcare) equilibrated in 20 mM HEPES–NaOH pH 7.5, 150 mM NaCl, 5 mM DTT.

GST-tagged CHC proteins were purified using standard procedures and the GST tag cleaved on-column using TEV protease. The flow-through after TEV cleavage was further purified using anion exchange chromatography and size-exclusion chromatography as described for ClACCp constructs.

## Protein crystallization

The Aurora-A 122–403 D274N, C290A, C393A/TACC3 519–563 complex was concentrated to 16.5 mg/ml; 5 mM ADP/MgCl$_2$ was added to the complex and incubated on ice for 1 h. The complex was screened against a range of commercial crystallization matrices. Drops were laid down at a 1:1 ratio of complex:precipitant in MRC sitting drop plates using a Mosquito LCP crystallization robot (ttplabtech) and incubated at 18°C. Cubic crystals were produced using 0.1 M Bis–Tris pH 5.5, 2 M ammonium sulphate as the precipitant after 2–3 days. Crystals were flash-frozen in liquid nitrogen with the addition of 30% ethylene glycol to the mother liquor to act as a cryoprotectant.

Diffraction data were collected from a single crystal at Diamond Light Source (Oxford, UK) on beamline I04-1. Autoprocessed data from *xia2* "3-daii" pipeline at Diamond Light Source were used for structure determination (Winter, 2010). Molecular replacement was performed in PHASER (McCoy *et al*, 2007) using the Aurora-A 122–403 C290A, C393A structure as a model (PDB 4CEG; Burgess & Bayliss, 2015). Clear difference density was observed for TACC3 519–563 and this was modelled in using Coot (Emsley & Cowtan, 2004). Subsequent rounds of iterative refinement were performed using Phenix (Adams *et al*, 2002) and Coot. MolProbity was used to determine structure quality (Chen *et al*, 2010).

ClACCp (10 mg/ml) was crystallized using the hanging drop vapour diffusion method. Diffraction-quality crystals were obtained by mixing 1.5 μl protein with 1.5 μl reservoir solution containing 0.1 M Tris pH 8.0, 0.12 M NaCl, 8% w/v polyethylene glycol 6000, 25% ethylene glycol. Crystals were flash-frozen in liquid nitrogen upon harvesting without additional cryoprotectant.

Diffraction data were collected at Diamond Light Source (Oxford, UK) at beamline I03. Autoprocessed data were used for molecular replacement using PHASER (McCoy *et al*, 2007) with CHC 1–494 (PDB 1BPO; ter Haar *et al*, 1998) as the search model. Four molecules were located in asymmetric unit. Subsequent iterative

refinement cycles were performed using Buster (Blanc *et al*, 2004), Phenix (Adams *et al*, 2002) and Coot (Emsley & Cowtan, 2004). MolProbity was used to assess geometry (Chen *et al*, 2010).

## Biochemical assays

Radioactive $^{32}$P-ATP kinase assays were performed as described previously (Burgess *et al*, 2015). In brief, 0.625 μM Aurora-A$^{M3KD}$ WT or selected mutant was incubated with 0.25 mg/ml MBP alone and in the presence of 5 μM TACC3$^{N-ACID}$ WT or mutant for 10 min. Reactions were terminated by the addition of 2% (v/v) orthophosphoric acid. Samples were spotted onto P81 Whatman paper, washed extensively with 0.2% (v/v) orthophosphoric acid and incorporation of radioisotope measured by scintillation counting.

Autophosphorylation assays were performed as for radioactive $^{32}$P-ATP reactions using Aurora-A 122–403 co-expressed with lambda phosphatase for 1 h at room temperature with His-Aurora-A D274N as a substrate in replace of MBP. Reactions were resolved by SDS–PAGE and subject to Western blotting using an α-phospho-T288 Aurora-A antibody (CST 2914, 1:2,000) as per the manufacturer's recommendation.

Aurora-A/TACC3 fluorescence polarization assays were performed as stated (Richards *et al*, 2016) using FAM-TACC3 peptides (PPR, Cambridge; Almac, Craigavon). Binding assays were performed in triplicate. FP signal was fit to a one-site total binding model in Prism 6 (GraphPad). Results are reported as $K_d \pm$ SE.

For CHC-TACC3 FP assays, CHC proteins were serially diluted in FP buffer (20 mM HEPES–NaOH pH 7.5, 200 mM NaCl, 5 mM DTT). FAM-labelled TACC3 peptide was added to a final concentration of 50 nM and incubated for 30 min at room temperature to allow binding to reach equilibrium. Fluorescence polarization measurements were made using a Victor X5 plate reader (Perkin Elmer) at 25°C with excitation at 490 nm and emission at 535 nm. Data analysis was performed as described above.

## NMR spectroscopy

Side chain assignments of TACC3 519–563 & 519–540 were obtained from $^{15}$N/$^{13}$C-labelled samples expressed as described before (Burgess *et al*, 2015) in 20 mM KPO$_4$ pH 7.0, 50 mM NaCl, 1 mM DTT, 0.02% (w/v) sodium azide at a protein concentration of 300 μM. 3D H(CCCO)NH, (H)C(CCO)NH, 13C HCCH-TOCSY, 13C NOESY-HSQC and 13C HSQC spectra were recorded for these peptides alone and in complex with Aurora-A 122–403 D274N on a Bruker Avance spectrometer operating at a 1H frequency of 700 MHz and a temperature of 298K. Spectra were recorded and processed using Bruker TopSpin™ 3.2 software (Bruker Biospin AG: Fällanden, Switzerland) and analysed using CCPN analysis.

For analysis of TACC3 549–570 helicity, synthetic peptides were weighed in to give a sample concentration of 2 mM in 20 mM Tris pH 7.2 and 50 mM NaCl. Spectra were recorded at a temperature of 290 K on a Bruker Avance-Neo 600 MHz spectrometer equipped with a prodigy cryoprobe. Peptides were assigned using 2D NOESY, TOCSY and 13C HSQC spectra recorded with watergate and gradient coherence selection, respectively. Assignment and chemical shift analysis was performed in CCPN analysis version 2.4. This software does not have standard random coil chemical shifts values for phospho-serine so it was not possible to calculate secondary chemical shift values for pS558 in the phosphorylated peptide.

## Generation of *TACC3* variants in HeLa cells by genome engineering

TACC-domain deletion and *TACC3*-F525A variants in HeLa cells were prepared by CRISPR-Cas9 method (Ran *et al*, 2013). See targeting strategy in Appendix Fig S1. Briefly, three guide RNAs with targeted cleavage sites near codon for F525 of human *TACC3* (Appendix Fig S1) were identified. To introduce the F525A point mutation, a single-stranded oligodeoxynucleotide (ssODN) was used as a template.

The guide RNAs were phosphorylated and cloned into BbsI-digested pX459 vector (Addgene; plasmid #48139). The guide RNA pairs (Sigma) were as follows:
gRNA1 (CACCGGCTCTCCTCTTTCAGCTCCA and AAACTGGAGCTG AAAGAGGAGAGCC),
gRNA2 (CACCGCAGGCAACGTACCCTCAGCG and AAACCGCTGA GGGTACGTTGCCTGC) and
gRNA3 (CACCGGCCAGGCAACGTACCCTCAG and AAACCTGAGGG TACGTTGCCTGGCC).

The sequence of the ssODN is as follows: GCCTTGAACTCTGCC AGCACCTCGCTTCCCACAAGCTGTCCAGGCAGTGAGCCAGTGCCCA CCCATCAGCAGGGGCAGCCTGCATTGGAGCTGAAAGAGGAG**AGCG CT**AGAGATCCAGCTGAGGgtacgttgcctggcacagacgtcacacacagtctgcccgg aggggatcccgtgagcacttgggcagctcgagacac with exon in upper and intron in lower case; AfeI restriction site in bold, which also includes the F525A mutation; the PAM sites for all three gRNAs were removed by introducing silent mutations (Appendix Fig S1).

Transfections were carried out using Lipofectamine 3000 (Invitrogen) and following manufacturer's guidelines. The day after transfection, cells were subjected to antibiotic selection using 0.5 μg/ml puromycin (Sigma) for 4 days. Surviving cells were serially diluted to one cell per well in 96-well plates. Single clones were expanded and screened by PCR amplification (Phusion, NEB) of the targeted genomic region (FP1-intron3: TGCCCCAGCCTCCACATTTGAAG and RP1-intron5: GGTTCAGGCTTCACATTTCATTATCAAG) (Appendix Fig S1) followed by restriction digestion with AfeI.

For promising clones, total RNA was isolated (QIAGEN RNeasy kit) with SuperScript III reverse transcriptase (Invitrogen). Using cDNA as template, the targeted region of *TACC3* was PCR-amplified (Phusion, NEB) (FP2-exon3: GAGTGACACCCGCCTCTGAGACCC TAG and RP2-exon8: CGCGGACGTCCTGAGGGAGTCTC). The PCR product was cloned into pJET1,2/Blunt (CloneJET; ThermoFisher scientific), and a minimum of 15 bacterial colonies was sequenced (Appendix Fig S1).

Western blot analysis was carried out using rabbit polyclonal antibody raised against aa 73–265 of human TACC3 (Gergely *et al*, 2000). TACC-domain deletion mutants were identified as a by-product of the CRISPR knock-in genome engineering.

## Cell culture and synchronization

HeLa WT, ΔTACC and F525A cells were maintained in Dulbecco's modified Eagle medium (DMEM) with GlutaMAX (Invitrogen)

supplemented with 10% heat-inactivated FBS, 100 U/ml penicillin, and 100 μg/ml streptomycin at 37°C in a 5% $CO_2$ atmosphere. Cell synchronization was achieved by a double-thymidine block. Cells at 30% confluency were treated with 2 mM thymidine in DMEM supplemented with 10% FBS for 18 h, released in growth media for 9 h, before addition of 2 mM thymidine for a further 17 h. Synchronization was confirmed by flow cytometry.

For RNAi and re-expression of GFP-tagged proteins in HeLa cells, siRNA and pBrain plasmids were used. Briefly, for TACC3, cells were transfected with siRNA on day 0, transfected 10 h later with pBrain plasmid and analysed on day 2. For CHC, cells were transfected with siRNA on day 0, transfected with pBrain plasmids on day 2 and fixed on day 4.

### Proximity ligation assay and microscopy

For PLA assays, cells grown on acid-etched glass coverslips were fixed with ice-cold methanol and incubated at −20°C for at least 15 min. Cells were rehydrated with phosphate-buffered saline (PBS) and blocked with 1% bovine serum albumin (BSA) in PBS for 30 min before incubation at room temperature for 1 h with the appropriate antibody diluted in 3% BSA in PBS. Primary antibodies used were mouse Aurora-A (Sigma, 1:500), rabbit TACC3 (Abcam, 1:200), and rabbit TPX2 (Novus biologicals, 1:500). A Duolink kit (Olink Bioscience, Uppsala, Sweden) was used according to the manufacturer's guidelines. Briefly, coverslips were incubated with secondary antibodies conjugated with oligonucleotides, PLA probe anti-mouse MINUS and PLA probe anti-rabbit PLUS, diluted in 3% BSA in PBS, for 1 h at 37°C inside a humidified chamber. After washes, cells were incubated at 37°C for 30 min with ligation mixture containing ligase. Following this, an amplification solution containing polymerase was incubated on the coverslips for 100 min at 37°C. Imaging was performed on a Zeiss LSM880 upright confocal microscope using a 40× oil objective (numerical aperture, 1.4). Z-stacks comprising 12–15 0.45 μm sections were acquired using Zen 2.1 software (Zeiss). Quantification of PLA dots was done using Imaris 8.3.0 software (Bitplane).

Primary antibodies used to analyse HeLa cell lines were TACC3 (against human TACC3 residues 73–265, 1:1,000, rabbit (Gergely *et al*, 2000); α-tubulin (Sigma-Aldrich, Dm1α, T9026, 1:1,000, mouse); Cas9 (Cell Signaling, 14697, 1:1,000, mouse). For α-tubulin staining, cells were fixed with 4% formaldehyde (Fisher Scientific) in PBS containing 0.01% Tween 20 (Promega) for 12 min at 37°C followed by permeabilization with PBS + 0.5% Triton X-100 (Acros Organics) at 37°C for 10 min. PBS + 5% BSA (Sigma-Aldrich) was used for blocking followed by incubation in primary antibodies for 1 h at 37°C for each step. After three washes of 5 min each in PBS + 0.1% Tween 20, cells were incubated with 1:1,000 diluted goat anti-rabbit Alexa Fluor 488 and goat anti-mouse Alexa Fluor 555 (Life Technologies) for 1 h at 37°C. After washing as described above, DNA was stained with 1 μg/ml Hoechst 33258 (Sigma-Aldrich) in PBS for 10 min followed by mounting in ProLong Diamond anti-fade (Invitrogen).

Fixed cells that were mounted in anti-fade medium were imaged on a scanning confocal microscope (Leica SP5) with 60×, 1.4 NA objective (Leica). Images presented are 3D projections of z-sections

taken every 0.5 μm across the cell. Images of any individual figure were acquired using the same settings and were imported into Volocity 6.3 (Perkin Elmer) or Photoshop CS6 (Adobe). Live cell imaging of HeLa WT TACC3 and variants was carried out on Zeiss Widefield Cell Observer Live Cell imaging system equipped with heating/$CO_2$ units using Plan-Apochromat 40×/0.95 NA Korr Ph3 M27 dry objective. Cells were seeded at 40,000 cells/well in to μ-slide 8-well dish (Ibidi, 80826); 30 min before imaging, the cells were washed once with FluoroBrite DMEM (A18967, Life Technologies) supplemented with 10% heat-inactivated FBS (Life Technologies) and 2 mM L-glutamine (Life Technologies) followed by the addition of 300 μl/well of the same medium containing 0.3 μM SiR-DNA (SC007, SPIROCHROME). Images were acquired in brightfield and Cy5 settings, every 5 min for 24 h with a step size of 3 × 1.5 μm. The cell cycle timings were analysed using ZEN 2 lite software. Statistical analyses and preparation of graphs were performed using Prism7 (GraphPad Software, Inc.). Numbers of experimental repeats (*n* values) are reported for each dataset in figures and figure legends.

For immunofluorescence studies on the CHC/TACC3 interface mutants, cells were fixed with PTEMF (50 mM PIPES, pH 7.2, 10 mM EGTA, 1 mM $MgCl_2$, 0.2% Triton X-100, and 4% paraformaldehyde) for 15 min at RT, and then permeabilized (PBS with 0.1% Triton X-100) for 10 min. Cells were blocked (PBS with 3% BSA, 5% goat serum) for 30 min, and then incubated for 1 h with the specified primary antibodies: mouse anti-CHC (X22, Affinity BioReagents, 1:1,000), rabbit anti-alpha tubulin (Thermo, PA5-19489, 1:1,000), mouse anti-TACC3 (AbCam, ab56595, 1:1,000). Secondary antibodies labelled with Alexa Fluor 568 or Alexa Fluor 647 from Life Technologies (diluted 1:500) were used along with DAPI. Images were taken using a Nikon Ti epifluorescence microscope with 100× objective (1.4 NA) and a Coolsnap Myo camera (Photometrics; Tucson, AZ). Equal exposure settings and scaling were used between conditions.

For spindle enrichment analysis, mean pixel density in two 10 × 10 pixel ROI was measured either over the spindle (away from poles), in the cytoplasm or in a non-cell region of background, using ImageJ. Background was subtracted from the spindle and cytoplasm measurements, and spindle fluorescence was then divided by cytoplasm fluorescence. Sample sizes 48–53 cells (TACC3) and 36–51 cells (CHC) over two experiments were analysed per condition. Ratios were analysed from cells with comparable fluorescence. Differences in spindle recruitment were evident at all expression levels. IgorPro was used to find and plot the ratio of spindles to cytoplasmic fluorescence intensity per cell. Images were cropped in ImageJ, and figures were assembled in Adobe Illustrator. Some images were contrast adjusted for presentation.

### Measurement of TACC3 volume at spindle pole regions

Quantification of the volume of TACC3 in spindle pole areas was done using Imaris 8.3.0 software (Bitplane). Briefly, maximum intensity projections of telophase cells were generated from z-stacks imaged using identical confocal settings and opened in Imaris. Surfaces were generated for voxels with intensities > 50, and volumes at spindle poles were selected and recorded individually.

## Data availability

The structure coordinates from this publication have been deposited to the RCSB PDB database (www.rcsb.org) and assigned the identifiers, 5ODS and 5ODT.

**Expanded View** for this article is available online.

## Acknowledgements

We thank the beamline scientists of Diamond I04-1 and I03 for their assistance with data collection. The work was funded through CRUK grants (C24461/A12772 and C24461/A23303 to RB, C14303/A17043 to FG and C25425/A15182 to SJR), BBSRC grant (BB/L023113/1 to RB and MP). FG acknowledges support from NIHR Cambridge Biomedical Research Centre, the University of Cambridge and Hutchison Whampoa Ltd. We thank Natarajan Kannan and Ruan Zheng for their insights into kinase interactions at the N-lobe cap, Emanuele Paci for his insights into helical peptides and Sheena Radford and John Ladbury for their comments on the manuscript.

## Author contributions

SGB, MM, SS, NJ, CG-C, MWR, NH-D, and MP designed and performed experiments. JWC and EJK contributed to the design of experiments. RB, FG, and SJR supervised experimental work. RB, FG, SJR, and SGB wrote the paper. All authors contributed to data analysis and commented on the manuscript.

## Conflict of interest

The authors declare that they have no conflict of interest.

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
