## [Review Process File · The EMBO Journal]

Mitotic spindle association of TACC3 requires Aurora-A-dependent stabilization of a cryptic α -helix

Selena G Burgess, Manjeet Mukherjee, Sarah Sabir, Nimesh Joseph, Cristina Gutiérrez-Caballero, Mark W Richards, Nicolas Huguenin-Dezot, Jason W Chin, Eileen J Kennedy, Mark Pfuhl, Stephen J Royle, Fanni Gergely & Richard Bayliss

Review timeline:

Submission date:	1 August 2017
Editorial Decision:	5 September 2017
Revision received:	20 December 2017
Editorial Decision:	24 January 2018
Revision received:	1 February 2018
Accepted:	2 February 2018

Editor: Hartmut Vodermaier

Transaction Report:

1st Editorial Decision

5 September 2017

Thank you for submitting your manuscript on the structural basis of TACC3 interactions for our editorial consideration. It has now been seen by three expert referees, whose comments are copied below for your information. As you will see, all reviewers generally appreciate the importance of this investigation and the overall interest of your findings. At the same time, they raise a number of substantive concerns that would need to be addressed prior to publication. Given that these issues appear to be mostly of technical or presentational nature and may in part be straightforward to clarify and/or experimentally strengthen, I would nevertheless like to give you an opportunity to respond to the referees' comments by way of a revised version of the manuscript.

Thus, should you be able to satisfactorily address the criticisms raised by the three reviewers, we should be happy to consider this work further for publication in The EMBO Journal. Please note, however, that it is our policy to allow only a single round of major revision, making it important to carefully respond to all points raised at this stage - therefore, please do not hesitate to get back to me with any questions/comments you may have regarding the referee reports already during the early stages of your revision. We might further discuss possible extension of the revision period (beyond the regular three months), during which time the publication of any competing work elsewhere would have no negative impact on our final assessment of your own study.

Please refer to the sections below for additional information on preparing and uploading a revised manuscript.

Thank you again for the opportunity to consider this work for The EMBO Journal, and I look forward to hearing from you in due time.

 REFEREE REPORTS

Referee #1:

This study presents novel structural, biophysical, and functional data to characterize important mechanisms underlying how the TACC3 protein is phosphorylated by the Aurora-A kinase and how phosphorylated TACC3 binds the Clathrin heavy chain (CHC). These protein interactions are critical for proper microtubule dynamics and mitotic spindle stability. The importance of the system for proper cell division and the general implications of the conclusions for phosphorylation-based signaling are significant, and the work will be of interest to a broad EMBO readership.

The work contains two distinct structures (Aurora-A-TACC3 complex and TACC3-CHC complex) that are each complimented by a set of complimentary biophysical and functional experiments. While the manuscript thus feels divided into two independent stories, they are connected, and it is reasonable to publish them together for greater impact. The study is logical and for the most part well designed and articulated. In the case of the Aurora-A-TACC3 complex, structural conclusions are rigorously supported by biophysical measurements and cell biology. An impressive example is the care given to identify the relevant TACC3-AuroraA interface in the crystal structure. A significant issue is the quality of the structural and supporting data for the TACC3-CHC complex. As discussed in detail in points 6-8 below, as presented, it is not convincing that the structural model has been built accurately and the interface identified correctly. This main concern and several others need to be addressed before the study is suitable for publication.

1) In Figure S1 panel B, there is not enough information provided to assess whether the gel filtration data support the proposed equimolar stoichiometry for Aurora-A and TACC3. Although the shifted chromatogram clearly shows a complex being formed, without comparison to molecular weight standards, there is no way to know what the apparent molecular weight of the peak is at the lower retention volume. Even with standards, it is unlikely one could differentiate the number of small TACC peptides that may be in the complex. A more accurate method like MALS or analytical ultracentrifugation would be needed. However, the fact that there is a clear heterodimer with one copy of each molecule in the asymmetric unit is strong support for the proposed stoichiometry. The solution stoichiometry measurement should be strengthened, or the data should be removed and stoichiometry assumed from the crystal structure.

2) Errors in the binding constant measurements from the FP assay in Figure 2 should be reported. In the caption and methods it is noted that results are reported +/- SE, but they are not indicated in the main text. Also, it may be useful to note the Kd measurements for each mutant in the legend in Fig. 2C and 2D (as in Fig. 4 for example).

3) The Kd measured for TACC3 522-536 by FP is compared to previous measurements using different approaches (MST or NMR) of TACC3 N-ACID to conclude that the 522-536 region is responsible for most of the affinity. It would be a more compelling comparison if the TACC3 N-ACID affinity were also measured by FP (using competition with the labeled 522-536 peptide for example) or if TACC3 522-536 affinity were measured by MST or NMR. It is common that these different approaches give variations in affinity so a comparison using the same approach is a much better controlled experiment.

4) The authors should clarify whether the observed TACC3 interaction with Aurora-A is "activating" because it is a docking interaction that enhances substrate capture, because it allosterically activates the kinase, or both. It seems that the authors conclude that both mechanisms are relevant, for example bottom of page 9: "Aurora-A recognizes TACC3 through a docking motif to a site of allosteric regulation". Or in the discussion on page 19, the section is titled, "A site for substrate docking and allosteric regulation". However, it is not clear what the evidence is that this peptide allosterically activates the enzyme, i.e. induces a structural change that enhances substrate binding or the catalytic step. The structure with bound peptide is in an inactive conformation. Although, as pointed out, the inactive conformation may reflect the requirement for activation loop, it begs the questions is TACC3 an allosteric activator and how? There are several enzymatic assays that could be used to distinguish whether the substrate interaction enhancement is from docking or

allosteric activation of catalytic activity. For example, a good experiment is to test whether a peptide with a phosphor-acceptor site mutation still binds and activates phosphorylation toward a peptide that has a docking site mutation. If the authors are going to claim that TACC3 is an allosteric activator, this should be supported with experiments here or with reference to previous experiments.

If there is sufficient evidence of allosteric activation, it is surprising the authors make no attempt to explain the mechanism using the structure. Interesting and informative comparisons are made to other kinases, but there is no comparison of the unbound and TACC3-bound Aurora-A structures. Are there any structural changes that could be understood as priming for activation even if the C-helix is not in the right conformation?

5) Considering that the function of the TACC3-CHC association is for stabilization of fibers and cross-linking, it is surprising that the affinity of the phosphorylated peptide is so weak. Are there data (here or already published) that implicate another interaction between other domains in these proteins that may lead to tighter association or is it known that this association is sufficient? This point should be addressed, particularly as the phenotypes of the mutants for co-localization in Fig. 6 are weak.

6) The x-ray diffraction data collected for the phosTACC3-CHC complex crystals are not of sufficient quality to fully support the model and conclusions. Although diffraction data are processed and used in refinement to 3.1 Å, the statistics in Table 1 suggest that the higher resolution data are not contributing much signal. In the outer shell: $I/\sigma = 1.2$, redundancy = 3.4, $R_{\text{merge}} = 123\%$ (this is noted incorrectly as 1.23%) suggest intensities have not been precisely measured. The mean B-factor for the protein (124.5 Å²) and the Wilson B-factor (99.33) are quite high and are reflective of disorder in the crystal. Also, why is the R_{free} (22.08 %) less than the R_{work} (26.9%)? This is either a typo or some nonconventional aspect of the refinement needs to be described. Consistent with these statistics and of greater importance, the quality of the electron density in the sample simulated annealing omit map in the Sup Fig. is weak, does not reflect the resolution expected for high quality data to 3.1 Å, and causes concern about whether sidechains were properly positioned in the helix. In the absence of collecting high quality data, which may understandably be challenging, the authors should take care to describe how the model was built and validated. What other markers besides the phosphoserine were used to anchor the sequence assignment? Does adding a model for the helix improve the R_{free} ? Were other possible models with different registry built and refined, and how do they comparatively affect the R_{free} ? How do the four different helix models in the asymmetric unit compare, and was NCS used for refinement? Beyond the general concern that the model may not be accurate, one specific example of a claim that is not supported by the data is that pS558 forms intramolecular salt bridges with R555 and K562. There is no unbiased electron density for either basic sidechain in the simulated annealing omit map that would support these interactions existing.

7) The accuracy of the helical model is especially important because of the interesting conclusion that the phosphoserine does not contact CHC but instead promotes association by stabilizing the helical conformation. While this has been observed in other systems, observing no interactions with the phosphate is rare and the phosphate is usually at the N-terminus of the helix, where it stabilizes the dipole and caps amide hydrogens (see PMID: 9413984 for the likely most well known case). For example, the solution NMR data in Fig. 5D for the unbound peptide, suggests that the pS558 is closer to the N-terminus of the helix than it is in the model built from the diffraction data (significant CSI less than -0.1 start just after pS558). Do the authors conclude that the conformation changes upon binding?

8) There are no biochemical or functional data that convincingly support the model of the TACC3 interface. All of the binding data in Fig. 5G were performed on mutants of TACC3, which is a short 21-mer peptide, so many residues would be expected to show some importance regardless of how it binds. All of the mutations give a similar effect except K562, and this flat structure-activity profile together with weak affinity suggest there is not much specificity in the interaction being probed. No measurements are made of mutations to the CHC domain based on the structure, but it would be considerably more compelling to probe mutations to both sides of the interface (as in Fig. 2 for the TACC3-Auror-A structure). In the functional assay, it does not appear that any of the mutations to CHC affect the rescue of TACC3 co-localization (only its own localization to the spindle), so it is not conclusive that these mutations are influencing the association in cells.

9) In Figure 6, although mentioned in the figure caption, markings indicating statistical significance do not appear on the actual data plots. Also, the difference between "kdEV" and "cEV" is not explained in the main text or figure legend.

Referee #2:

Clathrin and TACC3 proteins form a spindle-associated complex in mitosis that is necessary for microtubule stabilization and chromosome alignment. Aurora A kinase regulates formation of this complex, as its interaction with, and phosphorylation of, TACC3 is required for TACC3 and clathrin heavy chain (CHC) to interact and be recruited to mitotic spindle. In this study the authors investigate the structural basis of these interactions. The authors previously nicely demonstrated that TACC3 binds and activates Aurora A, with extensive analyses showing that the F525 residue of TACC3 is critical for the interaction/activation (Burgess et al., PLoS Genetics, 2015). In that study, they showed that mutation of this residue leads to less phosphorylation of TACC3, and thus less interaction with CHC and less recruitment of TACC3/CHC to the spindle. There they also showed that this mutation in DT40 cells, unlike inhibition of the TACC3-CHC interaction via mutation of a critical phosphorylation site, oddly led to cells building a spindle and exiting mitosis faster, without any apparent effect on chromosome alignment or segregation. They speculated that TACC3-specific activation of Aurora A has a minor yet specialized function (as opposed to the major TPX2-mediated activation of Aurora A).

In the current study, the authors present the structural basis for Aurora A interaction and activation by TACC3, and how this facilitates TACC3 interaction with CHC. Crystallographic and mutational studies of interacting Aurora A and TACC3 fragments confirm their previous NMR results that the F525 residue is central to the interaction. The Aurora A residues involved are also determined. The authors again introduce the F525A mutation into cells, this time HeLa cells, and again show that this causes reduction of TACC3-Aurora A interaction and less TACC3 on spindles. Unlike in DT40 cells, however, in HeLa cells this mutation leads to slower (not faster) mitoses, with delays both in prometaphase and telophase. It is suggested that the latter may be due to a TACC3-specific Aurora A function in later mitosis, based on this observed delay and differential localization of TACC3-Aurora A and TPX2-Aurora A complexes. The reason of the delay or consequence for the cells is not interrogated.

In the second part of this study, the authors determine the structural basis for interaction of CHC with phosphorylated TACC3. Again through crystallization and mutational analysis they define the residues on both TACC3 and CHC required for their interaction. Interestingly, the phosphorylation site on TACC3 does not appear to be involved in the direct interaction. Combining their structural/biochemical work, the authors are able to present a nice model of how TACC3 interaction with Aurora A stimulates its activity, leading to phosphorylation of TACC3 itself at S558, which in turn stabilizes a helix structure that allows interaction with CHC. The authors finally introduce mutations in the residues where TACC3 and CHC interact into cells to assay for their recruitment to the spindle. They claim these mutations significantly restrict CHC and TACC3 interaction and their localization in cells, although the data presented for this part as a whole is not that convincing.

Overall, the structural and biochemical studies presented here of TACC3 interaction/ activation of Aurora A and pTACC3 interaction with CHC are interesting, and given some clarification of presented data, provide an advancement in how we understand how this complex is established during mitosis, which would be important for the field and of general interest. The model proposed by the authors is compelling. On the other hand, the experiments in cells as presented unfortunately do not add much insight or clarity into either the role of TACC3-specific activation of Aurora A nor whether disruption of the TACC3-CHC interface alone is sufficient to abolish the formation of this complex on spindles. My feeling is that the structural/biochemical aspects of this paper have potential to warrant publication in EMBO Journal on their merits alone; however, the cellular analyses require significant revision/clarification prior to publication.

Major Comments:

1. Biochemical interaction studies.

The authors should provide the concentrations of proteins used in their assays, as this can be important for interpretation. For example, what were the concentrations of proteins used in SEC in Fig S1 to used to predict stoichiometry? Are these tagged constructs?

What were concentrations used in Fig S2B for kinase activity assays, that they may be compared to the established K_d of the relevant proteins? Also, it appears that the Aurora A mutations, including R151A, decrease the basal activity of the Aurora protein alone, potentially to a similar extent as when TACC3 is present. Thus, shouldn't this data be presented as a fold stimulation by TACC3 for each mutant?

2. Analysis of F525A mutants

The PLA signals on the spindle look beautiful. It would be nice to see a control showing that there is no signal in the assays from using the TACC3 antibody alone (the TACC3 K.O. line result nicely rules out a signal from the Aurora A antibody alone).

The authors conclude from PLA analysis that AurA-TACC3 and AurA-TPX2 have similar (not strictly "colocalization") localizations until late anaphase, where AurA-TACC3 is more at spindle poles, and AurA-TPX2 is more at central spindle/midbody. I don't dispute this, but it could be explained simply from the differential localization of TPX2 and TACC3 at this point of mitosis (TACC3 is more at spindle poles and TPX2 is more at central spindle). Is this assay telling us more? It would be thus helpful if this figure included the localization of all TACC3, TPX2, and AurA at the relevant time point in mitosis for comparison.

As mentioned above the metaphase phenotypic observations of the F525A mutant TACC3 (50% less interaction with AurA via PLA and reduction in localization to metaphase spindles) is consistent with their characterization of this mutation in DT40 cells where they additionally showed this mutant is less phosphorylated. What is new in this figure is the decreased localization/AurA-interaction of the mutant protein at the spindle poles that takes place at anaphase/telophase, which forms part of their argument for an anaphase/telophase function of TACC3-Aurora A. It would therefore be nice to see a clearer image of this localization defect with some sort of quantification. Furthermore, it would be informative to know if this localization phenotype is present in the DT40 cells with the same mutation, and if so, whether they display a delay in late anaphase as well.

In the text it is stated that Δ TACC3 and F525A mutant cells go slower from NEBD to anaphase. However, this data is not shown, nor the magnitude of the defect indicated. Please show this. While this is contradictory to the DT40 cells results, it would make more sense based on what is known about removing TACC3/CHC from spindles.

The authors need to be clear about reconciling their reported phenotypic differences of the F525A mutation in DT40 cells vs HeLa cells. This should be addressed in the discussion.

3. Analysis of CHC and TACC3 interaction interface mutants

The authors show that the mutating the residues in TACC3 observed contacting CHC from structural analysis significantly weaken their interaction (Fig 4G). No biochemical interaction analysis of the corresponding residues identified in CHC was shown. Both sets of mutations were introduced into cells to measure TACC3 and CHC localization to spindles. However, from this analysis I am not convinced that these mutations are of "critical importance" to localizing CHC and TACC3 as is claimed.

Fig 6A: From the images provided it does not appear by eye that there is much less TACC3-GFP nor CHC on the mutant spindles. The quantification data does look convincing, although if cytoplasmic signal was simply elevated after spindle association saturated due to overexpression, this could give misleading results based on the quantification method. Is the absolute intensity of TACC3-GFP over the entire spindles in mutants less than that of WT? The authors state in methods that "some images were contrast adjusted for presentation". Is this part of what's going on here? The authors should state which images are "contrast adjusted", and have

the same contrast settings for all images to be compared in a figure. Also, two great controls for these assay conditions would be a) the TACC3 S558A mutation, which has been shown by several groups to dramatically reduce TACC3 recruitment to spindles, and b) the authors' Δ TACC3 line, to show how much CHC remains on the spindle in this condition.

Fig 6B: The same quantification/presentation concerns as above for 6A applies here. Also, there is no statistical analysis of the quantification of TACC3 and CHC on spindles. Please report significance values (this applies to 6A as well). It really does not appear by eye that TACC3 localization is affected here. From the quantification, it does not appear that the WT CHC construct is recruiting any more TACC3 to spindles. It is way less than the empty vector control.

Overall the results presented lead me to believe that there could be two possible explanations for the apparently subtle effect seen here: either the introduced mutations do not weaken the CHC-TACC3 interaction enough to significantly impact cellular interaction (while significant, the *in vitro* data is not dramatic), and/or that in cells there are other factors present important for their spindle localization.

Minor Comments:

- 1) In my opinion the schematic of Fig 1A is too vague and speculative to be useful for the reader. It doesn't really represent the text where it is called in the introduction (to say ankle region of CHC and phosphorylated S558 of TACC3 interact and allow complex to bind MTs) - this is more clearly illustrated in the 1B diagram. It should also be clear that the illustration of a single clathrin triskelion molecule binding to two MTs in the last panel is highly speculative at this point if it is left in the graphic.
- 2) There should be a brief reasoning (or a reference) to why the "M3KD" mutations were introduced into Aurora A for structural work.
- 3) In the section on comparison of the TACC3-binding site in related kinases, why was Aurora B (or C) not compared?
- 4) There is a lack of citation of the earliest TACC3-CHC work. The first three papers showing the TACC3-CHC interaction, dependency on TACC3 pS558, and spindle localization dependencies (Lin et al JCB 2010, Hubner et al JCB 2010, Fu et al JCS 2010) are absent.
- 5) The description of the TACC3 F525A mutant cell line made via CRISPR/Cas9 should more accurately reflect that there could be a significant proportion of protein with the non-desired mutations. For example "a small proportion may contain the E519D and..." should be changed to "a proportion contains the E519D and". It is not possible to ascertain the relative amounts from the 15 clones analyzed that were picked from a bacterial transformation of cDNA cloned into vectors (and at least one chromosome will have the 10aa deletion and at least another will have the truncation).

Referee #3:

Burgess et al report an extensive structural, biochemical and cell-based study that elucidates the mechanism for the specific phosphorylation of the AurA substrate TACC3 and subsequent phosphorylation-driven interaction of TACC3 with clathrin heavy chain (CHC), an important process required to assemble cross bridges that stabilize mitotic spindle microtubules. Using crystallography the authors show that AurA recruits the TACC3 substrate through a specific and novel docking interaction on the N-terminal lobe of the kinase, and they validate this interaction by mutagenesis, binding assays, and NMR, as well as in cells. They also present an x-ray structure of an engineered CHC-TACC3 fusion protein that reveals how phosphorylation of TACC3 by AurA promotes the interaction with CHC by triggering a disorder-to-order transition of the TACC3 interaction segment. This interesting mechanism is also backed up by extensive mutagenesis, binding assay and NMR experiments.

Although the AurA kinase system has been heavily studied, and many *in vivo* interaction partners of

have been identified, surprisingly little is known about the molecular mechanisms underlying many of these critical processes. This study makes a major step forward in our understanding of the structural mechanisms underpinning physiological AurA-driven interactions in mitosis and is of high significance. The experimental approaches are very strong and the key assertions are all backed up by extensive experimental evidence both in vitro and in cells. The manuscript is very well written and frankly a pleasure to read. I strongly recommend publication of the paper essentially as is, although I would suggest the authors consider the comments raised below and make some minor revisions to the manuscript accordingly, which can be addressed without further review.

- 1) the authors do not extensively discuss the conformation of the AurA kinase domain in the TACC3 cocrystal structure, although they state it is in an inactive conformation. On the face of it, this is somewhat surprising given that TACC3 binding actually activates AurA. It is stated that the kinase α C-helix is distorted and the activation loop is in an inactive conformation because of the lack of phosphorylation, but other x-ray structures of unphosphorylated AurA show the kinase in the DFG-In state without a distorted α C-helix (e.g. 1MQ4), although they typically also show that the activation loop is not properly positioned for substrate binding. Can the authors describe the conformation of the kinase more fully (for instance is the kinase in the DFG-In or DFG-Out state?), put it into context with respect to the other available structures of AurA, and speculate as to why the kinase domain does not adopt a more active-like conformation? Is it possible that the other contacts with TACC3 in the lattice (sites 2 and 3) are influencing the conformation of the kinase domain?
- 2) The authors do not comment on how the TACC3 site 1:AurA docking interaction would position the downstream segment containing the S558 phosphorylation site relative to the catalytic site of the kinase. Presumably the ~20 intervening residues would be more than sufficient to span the distance, but it would be nice to have this explicitly mentioned.
- 3) It is unclear whether the kinase activity assays showing activation of AurA by TACC3 (Figure S2) were done with phosphorylated or unphosphorylated AurA. If activation of AurA by TACC3 requires activation loop phosphorylation (unlike activation by Tpx2) then this might help explain the inactive conformation of AurA observed in the x-ray structure (point 1 above). If so, perhaps the authors could make this point more explicit.
- 4) The authors state that the AurA/TACC3 interaction is conserved in drosophila, but do not comment on whether the interface visualized in their x-ray structure is also conserved across metazoans. Admittedly this may be hard to address given the high degree of sequence conservation in metazoan kinases.
- 5) Site 2 in the AurA:TACC3 structure looks superficially like the AurA:Tpx2 interaction. Can the authors comment on this if there is indeed any real resemblance?
- 6) It is not clear why the R555A/K556A double mutant was tested for disruption of the TACC3:CHC interaction as opposed to the single R555A or double R555A/K562A mutants. Can they clarify this point?
- 7) the crystallographic data table lists an Rfree value that is lower than the Rwork value for the CHC/TACC3 structure. This is probably a typo, as lower Rfree values are very rare and usually indicate incorrect assignment of the test set flags.
- 8) the authors cite Filappakopoulos et al 2008 for the Abl:SH2 NCAP interaction, but this was technically first shown by the Kuriyan group (Nagar et al Mol. Cell 2006). The authors also omit several other relevant references for the Tpx2 activation mechanism cited in the introduction, e.g. Zorba et al 2014, Cyphers et al., 2017.

1st Revision - authors' response

20 December 2017

We thank the reviewers for their supportive and helpful comments on the first draft of this manuscript. We have addressed their concerns in a point-by-point response below.

Referee #1:

1) In Figure S1 panel B, there is not enough information provided to assess whether the gel filtration data support the proposed equimolar stoichiometry for Aurora-A and TACC3. Although the shifted chromatogram clearly shows a complex being formed, without comparison to molecular weight standards, there is no way to know what the apparent molecular weight of the peak is at the lower retention volume. Even with standards, it is unlikely one could differentiate the number of small TACC peptides that may be in the complex. A more accurate method like MALS or analytical ultracentrifugation would be needed. However, the fact that there is a clear heterodimer with one copy of each molecule in the asymmetric unit is strong support for the proposed stoichiometry. The solution stoichiometry measurement should be strengthened, or the data should be removed and stoichiometry assumed from the crystal structure.

The reviewer is correct in that the gel filtration profiles only confirm complex formation and not stoichiometry. We take their suggestion to infer stoichiometry from the crystal structure and have amended the manuscript text to reflect this.

2) Errors in the binding constant measurements from the FP assay in Figure 2 should be reported. In the caption and methods it is noted that results are reported +/- SE, but they are not indicated in the main text. Also, it may be useful to note the K_d measurements for each mutant in the legend in Fig. 2C and 2D (as in Fig. 4 for example).

The suggested changes have been made to the text and figures.

3) The K_d measured for TACC3 522-536 by FP is compared to previous measurements using different approaches (MST or NMR) of TACC3 N-ACID to conclude that the 522-536 region is responsible for most of the affinity. It would be a more compelling comparison if the TACC3 N-ACID affinity were also measured by FP (using competition with the labeled 522-536 peptide for example) or if TACC3 522-536 affinity were measured by MST or NMR. It is common that these different approaches give variations in affinity so a comparison using the same approach is a much better controlled experiment.

We performed a FP binding assay between Aurora-A and FAM-TACC3 519-570, which is slightly longer than the TACC3^{N-ACID} (519-563) construct used for MST and NMR binding quantification. The resulting K_d was 3.0 ± 0.2 μM (Fig EV2A) and is in close agreement with the K_ds measured by MST and NMR (6-8 μM).

4) The authors should clarify whether the observed TACC3 interaction with Aurora-A is "activating" because it is a docking interaction that enhances substrate capture, because it allosterically activates the kinase, or both. It seems that the authors conclude that both mechanisms are relevant, for example bottom of page 9: "Aurora-A recognizes TACC3 through a docking motif to a site of allosteric regulation". Or in the discussion on page 19, the section is titled, "A site for substrate docking and allosteric regulation". However, it is not clear what the evidence is that this peptide allosterically activates the enzyme, i.e. induces a structural change that enhances substrate binding or the catalytic step. The structure with bound peptide is in an inactive conformation. Although, as pointed out, the inactive conformation may reflect the requirement for activation loop, it begs the question is TACC3 an allosteric activator and how? There are several enzymatic assays that could be used to distinguish whether the substrate interaction enhancement is from docking or allosteric activation of catalytic activity. For example, a good experiment is to test whether a peptide with a phosphor-acceptor site mutation still binds and activates phosphorylation toward a peptide that has a docking site mutation. If the authors are going to claim that TACC3 is an allosteric activator, this should be supported with experiments here or with reference to previous experiments.

We have previously shown in Burgess et al (2015) that phosphonull TACC3 is still capable of activating Aurora-A and we have added this citation to this part of the paper. To expand on how TACC3 activates Aurora-A, we have performed an additional kinase assay (Fig 2E) which shows TACC3 enhances Aurora-A autophosphorylation indicating TACC3 is an allosteric activator.

If there is sufficient evidence of allosteric activation, it is surprising the authors make no attempt to explain the mechanism using the structure. Interesting and informative comparisons are made to

other kinases, but there is no comparison of the unbound and TACC3-bound Aurora-A structures. Are there any structural changes that could be understood as priming for activation even if the C-helix is not in the right conformation?

We thank the reviewer for this helpful suggestion. Indeed, comparison of the TACC3-bound Aurora-A with other inactive structures suggests a potential mechanism for allosteric activation, that we missed in our initial analysis of the structure. This is shown in Figure 3C-E and the new text in the section “A model for allosteric activation of Aurora-A by TACC3”.

5) Considering that the function of the TACC3-CHC association is for stabilization of fibers and cross-linking, it is surprising that the affinity of the phosphorylated peptide is so weak. Are there data (here or already published) that implicate another interaction between other domains in these proteins that may lead to tighter association or is it known that this association is sufficient? This point should be addressed, particularly as the phenotypes of the mutants for co-localization in Fig. 6 are weak.

There probably are additional interactions that contribute to the association between these proteins in the cell, particularly in the context of microtubules. While the interaction between the ankle domain of CHC and the CID of TACC3 are necessary for complex formation and microtubule association, these domains are not the contact points with microtubules. We previously showed (in Hood et al., JCB 2013) that the TACC3-CHC complex associates with microtubules through the propeller domain of CHC and the coiled-coil (TACC) domain of TACC3, and that a chimeric protein formed through the fusion of these domains localizes to microtubules. A chimera between an alternative coiled coil protein and the CHC propeller does not bind MTs, and so our current model is that the propeller of CHC and coiled-coil of TACC3 must interact to generate a MT binding interface. We think it likely that these domains associate with each other and thereby increase the overall affinity of the TACC3-CHC interaction, but we have no data on this. Moreover, there are additional components of the complex (e.g. GTSE1, see Hubner et al. JCB 2010) that could contribute to complex assembly.

6) The x-ray diffraction data collected for the phosTACC3-CHC complex crystals are not of sufficient quality to fully support the model and conclusions. Although diffraction data are processed and used in refinement to 3.1 Å, the statistics in Table 1 suggest that the higher resolution data are not contributing much signal. In the outer shell: I/σ = 1.2, redundancy = 3.4, Rmerge = 123% (this is noted incorrectly as 1.23%) suggest intensities have not been precisely measured. The mean B-factor for the protein (124.5 Å²) and the Wilson B-factor (99.33) are quite high and are reflective of disorder in the crystal. Also, why is the Rfree (22.08 %) less than the Rwork (26.9%)? This is either a typo or some nonconventional aspect of the refinement needs to be described. *This was a typo – Rfree is 26.9%, Rwork is 22.08%*

Consistent with these statistics and of greater importance, the quality of the electron density in the sample simulated annealing omit map in the Sup Fig. is weak, does not reflect the resolution expected for high quality data to 3.1 Å, and causes concern about whether sidechains were properly positioned in the helix. In the absence of collecting high quality data, which may understandably be challenging, the authors should take care to describe how the model was built and validated. What other markers besides the phosphoserine were used to anchor the sequence assignment?

In recent papers we have tended to only briefly describe how we determined crystal structures, mainly because it seems most people are not interested in these details. We are happy to provide them here, and as an additional figure in the Appendix (Appendix Figure S3).

Molecular replacement gave clear positions for four CHC beta propellers and their associated ankle regions. Fitting the ankles to the density was a challenge because these helical repeats are flexible and were packed into different conformations in the crystal lattice. We also did not have an experimental structure for the most C-terminal repeats. Map generation in BUSTER helped to reduce model bias and the correct positions of these helices was then clear.

Based on the maps after we had finished our model of CHC (Appendix Figure S3), four features of the TACC3 peptide were clear:

- 1) *A region of alpha-helix*
- 2) *A region of non-helix, suggesting a very compact turn*

- 3) *A large side chain on the α -helix positioned at some distance from the surface of CHC (3 in the figure below)*
- 4) *Another large dense side chain, at the $i+5$ position relative to the first electron-dense side chain (4 in the figure below)*

We fit the model to the best register and realized that TACC3 Tyr560 could fit if we moved the sidechain of CHC Arg481 to form a cation- π stack (landmark feature 5). However, we recognize that, based on our data alone, we could not distinguish between similar residues (e.g. Phe vs Tyr). We considered alternative registers, based on the observation of 3 landmark sidechain features in the density (labelled 3,4,5 below).

FKESALRKQSLYLKFDPLLDRS	- map extends beyond C-terminus
FKESALRKQSLYLKFDPLLDRS	- Pro poor fit to SC density at 4
FKESALRKQSLYLKFDPLLDRS	=> final model
FKESALRKQSLYLKFDPLLDRS	- *Refined – see below.
FKESALRKQSLYLKFDPLLDRS	- Ala poor fit to SC density at 3
3 5 4	- positions of landmark features

With the helix in place, and the register of the sequence determined, the arrangement of the C-terminus became clear: the sidechain of Asp564 forms an intra-molecular H-bond with the mainchain N of Leu566, separated by Pro565. This arrangement fits the density.

Does adding a model for the helix improve the Rfree?

Yes, adding a model for the helix in all 4 chains improves Rfree by 1%

Were other possible models with different registry built and refined, and how do they comparatively affect the Rfree?

*Yes, other models were built, but none fit the density as well as the model in the manuscript. For example the register marked with * above was refined in chain E, resulting in Rfree of 27.14%. The final maps confirm a worse fit to the model, e.g. no density for the modelled pSer.*

How do the four different helix models in the asymmetric unit compare, and was NCS used for refinement?

The four helix models were built from a template derived from the first TACC3 peptide we built (chain E) and we used torsion-angle restraints in NCS. The models are similar and consistent with electron density, but the density is weaker in the other chains (see Appendix Figure S3C-F)

Beyond the general concern that the model may not be accurate, one specific example of a claim that is not supported by the data is that pS558 forms intramolecular salt bridges with R555 and K562. There is no unbiased electron density for either basic sidechain in the simulated annealing omit map that would support these interactions existing.

Here we recognize that our structural data on its own cannot support the claim that these salt bridges exist, and we provide additional data in support of our model. First of all, we have carried out binding assays using the R555A mutant, phosphorylated TACC3 peptide – this has a Kd for CHC binding similar to that of the unphosphorylated peptide. Furthermore, we have repeated and extended the NMR experiments to probe the helicity of the TACC3 in solution (see response to point 7 below).

7) The accuracy of the helical model is especially important because of the interesting conclusion that the phosphoserine does not contact CHC but instead promotes association by stabilizing the helical conformation. While this has been observed in other systems, observing no interactions with the phosphate is rare and the phosphate is usually at the N-terminus of the helix, where it stabilizes the dipole and caps amide hydrogens (see PMID: 9413984 for the likely most well known case). For example, the solution NMR data in Fig. 5D for the unbound peptide, suggests that the pS558 is closer to the N-terminus of the helix than it is in the model built from the diffraction data (significant CSI less than -0.1 start just after pS558). Do the authors conclude that the conformation changes upon binding?

The reviewer's comments prompted us to reanalyse the peptide samples. Further experiments showed that the two peptides were in solutions of different pH and that the pSer558 would have perhaps a charge of only -1. This was unacceptable. So we repeated the solution NMR experiments using freshly prepared stocks of newly-synthesized peptides in properly buffered solution. Analysis was expanded to include analysis of the alpha carbon atoms because their chemical shifts are less

influenced by charged side chains. The new data were collected and analysed by Dr Mark Pfuhl, an experienced NMR spectroscopist and author on the paper (whereas the previous data was collected and analysed by a less experienced researcher). The new data (in Figure 7) shows changes in proton and carbon chemical shifts that reflect increased helix propensity between aa553 and 563 of TACC3, consistent with our model of phosphorylation-induced helix stabilisation.

8) There are no biochemical or functional data that convincingly support the model of the TACC3 interface. All of the binding data in Fig. 5G were performed on mutants of TACC3, which is a short 21-mer peptide, so many residues would be expected to show some importance regardless of how it binds. All of the mutations give a similar effect except K562, and this flat structure-activity profile together with weak affinity suggest there is not much specificity in the interaction being probed. No measurements are made of mutations to the CHC domain based on the structure, but it would be considerably more compelling to probe mutations to both sides of the interface (as in Fig. 2 for the TACC3-Auror-A structure). In the functional assay, it does not appear that any of the mutations to CHC affect the rescue of TACC3 co-localization (only its own localization to the spindle), so it is not conclusive that these mutations are influencing the association in cells.

We were surprised by the relatively flat structure-activity of this set of peptides – and this is worthy of comment and we inserted additional text into the manuscript just after the data is presented “However, a wider range of altered CHC-binding properties in the TACC3 variant peptides should perhaps be expected and so further functional analysis of the interaction was required to increase confidence in the model of the TACC3/CHC interface.”

We agree that the mutations on both sides of the interface should be probed, and new data are included in the revised manuscript. Based on the structure, we generated mutations in the CHC surface that disrupt the interaction with the TACC3 CID peptide (New Fig. 6A,B). Strikingly, mutations in the pocket into which F563 of TACC3 binds reduce the affinity of interaction to >300 μ M, in the same range as the TACC3 F563A mutant binding to wild-type CHC. Mutation of R481 to glutamic acid achieves a similar reduction in binding, because it disrupts the electrostatic interaction with D564 of CHC and a cation- π interaction with Y560.

In the functional, cell-based assay, rescue of TACC3 spindle association with wild-type CHC is incomplete. We therefore do not draw any conclusions based on the TACC3 spindle association in the presence of the mutants, even though it appears to be reduced compared to the wild-type CHC rescue.

9) In Figure 6, although mentioned in the figure caption, markings indicating statistical significance do not appear on the actual data plots. Also, the difference between "kdEV" and "cEV" is not explained in the main text or figure legend.

The plots in this figure have been updated to include statistical significance. The terms “kdEV” and “cEV” have been removed from the figure, replaced with the descriptors “Control RNAi/GFP” and “CHC RNAi/GFP”.

Referee #2:

Major Comments:

1. Biochemical interaction studies.

The authors should provide the concentrations of proteins used in their assays, as this can be important for interpretation. For example, what were the concentrations of proteins used in SEC in Fig S1 to used to predict stoichiometry? Are these tagged constructs?

Based on comments from reviewer 1, we have removed statements regarding estimation of complex stoichiometry from the gel filtration profiles and have instead inferred this from the crystal structure. The constructs used were untagged.

What were concentrations used in Fig S2B for kinase activity assays, that they may be compared to the established K_d of the relevant proteins?

Assay conditions have been expanded in the methods section to include this information.

Also, it appears that the Aurora A mutations, including R151A, decrease the basal activity of the Aurora protein alone, potentially to a similar extent as when TACC3 is present. Thus, shouldn't this data be presented as a fold stimulation by TACC3 for each mutant?

Yes, we noticed the basal activities of the mutants were different and thank the reviewer for suggesting a more relevant method by which to display this data. The data has been reanalyzed as requested to reflect enhancement of each individual mutant on addition of TACC3.

2. Analysis of F525A mutants

The PLA signals on the spindle look beautiful. It would be nice to see a control showing that there is no signal in the assays from using the TACC3 antibody alone (the TACC3 K.O. line result nicely rules out a signal from the Aurora A antibody alone).

This control experiment is included in the revised figure EV4B.

The authors conclude from PLA analysis that AurA-TACC3 and AurA-TPX2 have similar (not strictly "colocalization") localizations until late anaphase, where AurA-TACC3 is more at spindle poles, and AurA-TPX2 is more at central spindle/midbody. I don't dispute this, but it could be explained simply from the differential localization of TPX2 and TACC3 at this point of mitosis (TACC3 is more at spindle poles and TPX2 is more at central spindle). Is this assay telling us more? It would be thus helpful if this figure included the localization of all TACC3, TPX2, and AurA at the relevant time point in mitosis for comparison.

The revised figure EV4 shows localization of TACC3 (C), Aurora-A (D) and TPX2 (E) at individual stages of mitosis. Indeed, the AurA-TACC3 and AurA-TPX2 PLA data are consistent with the localization of TPX2/TACC3. The localization of AurA does not track perfectly with either of these interaction partners – it is more concentrated at spindle poles – until cytokinesis when some AurA relocates to the central spindle with TPX2. The assay does not tell us more than this – our aim was to establish whether the F525A reduces the interaction with Aurora-A in mitotic cells.

As mentioned above the metaphase phenotypic observations of the F525A mutant TACC3 (50% less interaction with AurA via PLA and reduction in localization to metaphase spindles) is consistent with their characterization of this mutation in DT40 cells where they additionally showed this mutant is less phosphorylated. What is new in this figure is the decreased localization/AurA-interaction of the mutant protein at the spindle poles that takes place at anaphase/telophase, which forms part of their argument for an anaphase/telophase function of TACC3-Aurora A. It would therefore be nice to see a clearer image of this localization defect with some sort of quantification.

Quantification of WT-TACC3 and F525A-TACC3 localisation is included as new Figure 4 (and EV4).

Furthermore, it would be informative to know if this localization phenotype is present in the DT40 cells with the same mutation, and if so, whether they display a delay in late anaphase as well.

Yes, TACC3 in DT40 cells with the equivalent mutation (F543A) also shows reduced spindle localization, which we mention in additional sentences in the Discussion. Anaphase appeared normal in F543A DT40 cells, but we were unable to reliably measure the duration of telophase-cytokinesis in these cells due to their morphology.

In the text it is stated that Δ TACC3 and F525A mutant cells go slower from NEBD to anaphase. However, this data is not shown, nor the magnitude of the defect indicated. Please show this. While this is contradictory to the DT40 cells results, it would make more sense based on what is known about removing TACC3/CHC from spindles.

This data is included as Figure EV4H.

The authors need to be clear about reconciling their reported phenotypic differences of the F525A mutation in DT40 cells vs HeLa cells. This should be addressed in the discussion.

We have inserted the following text into the Discussion. "The reduced spindle localization of the F525A mutant in HeLa cells is similar to what we previously observed with the F525A equivalent mutation in chicken DT40 cells, but there is a critical difference in the mitotic timings between these cellular systems. The F525A equivalent mutation in DT40 cells shortened the duration of NEBD to anaphase onset by ~5 minutes, whereas F525A-HeLa cells exhibited a delay of ~35 minutes compared to single-cloned controls, suggesting increased reliance of spindle assembly in HeLa cells on TACC3-containing inter-microtubule bridges. Indeed, these bridges are likely to play a more central role in stabilizing human K-fibres that contain 20-40 microtubules, than DT40 K-fibres, which comprise only ~4 microtubules (Nixon et al, 2015; Ribeiro et al, 2009)."

3. Analysis of CHC and TACC3 interaction interface mutants

The authors show that the mutating the residues in TACC3 observed contacting CHC from structural analysis significantly weaken their interaction (Fig 4G). No biochemical interaction analysis of the corresponding residues identified in CHC was shown. Both sets of mutations were introduced into cells to measure TACC3 and CHC localization to spindles. However, from this analysis I am not convinced that these mutations are of "critical importance" to localizing CHC and TACC3 as is claimed.

Fig 6A: From the images provided it does not appear by eye that there is much less TACC3-GFP nor CHC on the mutant spindles. The quantification data does look convincing, although if cytoplasmic signal was simply elevated after spindle association saturated due to overexpression, this could give misleading results based on the quantification method.

Is the absolute intensity of TACC3-GFP over the entire spindles in mutants less than that of WT? The authors state in methods that "some images were contrast adjusted for presentation". Is this part of what's going on here? The authors should state which images are "contrast adjusted", and have the same contrast settings for all images to be compared in a figure. Also, two great controls for these assay conditions would be a) the TACC3 S558A mutation, which has been shown by several groups to dramatically reduce TACC3 recruitment to spindles, and b) the authors' Δ TACC3 line, to show how much CHC remains on the spindle in this condition.

The referee is correct that spindle recruitment ratios will be decreased in cells that have elevated cytoplasmic protein despite equal amounts of protein bound to the spindle. We check for this phenomenon by plotting spindle fluorescence versus cytoplasmic fluorescence and fit the data to assess if the spindle has become "maxed out" as the referee suggests. The ratios are constant over a range of expressions and all the conditions have similar fluorescence (with the exception of GFP alone). We prefer to present the spindle ratio as it is simple to understand. We have added a note to explain that the ratios observed occur at a range of expression values.

Instead of S558A we used our AARDS mutant (LL566,567AA) which we showed previously was equivalent to the S558A mutation. We have not tested the Δ TACC3 cell line because this comparison would be comparing TACC3 status and a further variable (cell line lineage) and so this control is not better than the RNAi control that we have performed.

Fig 6B: The same quantification/presentation concerns as above for 6A applies here. Also, there is no statistical analysis of the quantification of TACC3 and CHC on spindles. Please report significance values (this applies to 6A as well).

Statistical analysis of the data is provided in the revised version. Most of the mutations that disrupt the CHC-TACC3 interaction reduce the spindle localization of the mutant protein to ($p < 0.001$), using ANOVA with Tukey's post-hoc test.

It really does not appear by eye that TACC3 localization is affected here. From the quantification, it does not appear that the WT CHC construct is recruiting any more TACC3 to spindles. It is way less

that the empty vector control.

It is true that the rescue of TACC3 spindle localization with re-expression of WT CHC is incomplete. This is possibly due to poor expression of this large protein in HeLa cells.

Overall the results presented lead me to believe that there could be two possible explanations for the apparently subtle effect seen here: either the introduced mutations do not weaken the CHC-TACC3 interaction enough to significantly impact cellular interaction (while significant, the in vitro data is not dramatic), and/or that in cells there are other factors present important for their spindle localization.

The data show that the new mutations designed on the basis of the structure are equivalent to our previous "best in class" for disruption of this interaction and localization. However, the referee is right that there could be other factors important for spindle localization of CHC-TACC3. Our previous work suggested that there are sites on the N-terminus of clathrin and the TACC domain of TACC3 that are important for microtubule binding (Hood et al. JCB 2013). Both of these sites are distinct from the CHC-TACC3 interaction and one possibility is that CHC binds GTSE1, identified by Hubner et al. (JCB 2010) as a clathrin binding partner. Recent work suggests PI3KC2alpha is important for stabilizing CHC-TACC3 at the spindle (Gulluni et al. 2017 Cancer Cell). The experiments presented here are simply to test the CHC-TACC3 interaction and not to probe how this multiprotein complex interacts with microtubules. We have amended the final paragraph of the Results section (relocated to the penultimate section of the Results in the revised manuscript) to clarify this point – softening our language from the emphatic "is of critical importance" to say that the interface "contributes" to the localization of CHC and TACC3 in cells.

Minor Comments:

1) In my opinion the schematic of Fig 1A is too vague and speculative to be useful for the reader. It doesn't really represent the text where it is called in the introduction (to say ankle region of CHC and phosphorylated S558 of TACC3 interact and allow complex to bind MTs) - this is more clearly illustrated in the 1B diagram. It should also be clear that the illustration of a single clathrin triskelion molecule binding to two MTs in the last panel is highly speculative at this point if it is left in the graphic.

To improve the clarity of the paper, we have removed Fig 1A from the manuscript.

2) There should be a brief reasoning (or a reference) to why the "M3KD" mutations were introduced into Aurora A for structural work.

A description of the reasoning behind the M3KD mutant is in the Materials and Methods. Please see the 'Protein expression and purification' section.

3) In the section on comparison of the TACC3-binding site in related kinases, why was Aurora B (or C) not compared?

Aurora-B and C are virtually identical to Aurora-A in the residues that line the TACC3 binding site, which partially overlaps with the binding site for INCENP (Fig. EV3F,G). INCENP presumably outcompetes TACC3 for binding to Aurora-B, based on the localization of this kinase to kinetochores and not spindle microtubules. The potential for binding of Aurora-B/C to TACC3 may be an interesting topic for future studies.

4) There is a lack of citation of the earliest TACC3-CHC work. The first three papers showing the TACC3-CHC interaction, dependency on TACC3 pS558, and spindle localization dependencies (Lin et al JCB 2010, Hubner et al JCB 2010, Fu et al JCS 2010) are absent.

We have added these to the relevant positions in the introduction. We thank the reviewer for picking up on our omissions.

5) The description of the TACC3 F525A mutant cell line made via CRISPR/Cas9 should more accurately reflect that there could be a significant proportion of protein with the non-desired

mutations. For example "a small proportion may contain the E519D and...." should be changed to "a proportion contains the E519D and". It is not possible to ascertain the relative amounts from the 15 clones analyzed that were picked from a bacterial transformation of cDNA cloned into vectors (and at least one chromosome will have the 10aa deletion and at least another will have the truncation).

We have amended this statement in the paper. Assuming that there are two alleles of TACC3 in HeLa cells (HeLa are diploid for Chr 8), our data suggests that one allele has the correct point mutation, whereas the second allele has a 30bp deletion. Because this deletion is proximal to the splice junction, in addition to the 10aa deletion, this mutant allele is likely to produce the cDNA product that lacks exon5, but which does not yield a protein.

Referee #3:

1) the authors do not extensively discuss the conformation of the AurA kinase domain in the TACC3 cocystal structure, although they state it is in an inactive conformation. On the face of it, this is somewhat surprising given that TACC3 binding actually activates AurA. It is stated that the kinase aC-helix is distorted and the activation loop is in an inactive conformation because of the lack of phosphorylation, but other x-ray structures of unphosphorylated AurA show the kinase in the DFG-In state without a distorted aC-helix (e.g. 1MQ4), although they typically also show that the activation loop is not properly positioned for substrate binding. Can the authors describe the conformation of the kinase more fully (for instance is the kinase in the DFG-In or DFG-Out state?), put it into context with respect to the other available structures of AurA, and speculate as to why the kinase domain does not adopt a more active-like conformation? Is it possible that the other contacts with TACC3 in the lattice (sites 2 and 3) are influencing the conformation of the kinase domain?

The revised manuscript includes this analysis. Unphosphorylated Aurora-A has been captured with or without a distorted C-helix, depending on crystallization conditions used. 1MQ4 has an active-like conformation, but this might be due to the presence of a phosphate in the crystallization buffer that mimics phosphorylation. Indeed, we believe that the inactive state has a distorted helix, based on work by Zorba et al. (2014) and Cyphers et al. (2017), together with our own work Burgess et al. (2015). Our new analysis suggests that the TACC3 structure is an intermediate state – distorted C-helix, but DFG-in. However, our conclusions are cautious because of the potential for crystal packing interactions (such as site-2 and site-3) to influence the conformation of the kinase.

2) The authors do not comment on how the TACC3 site 1:AurA docking interaction would position the downstream segment containing the S558 phosphorylation site relative to the catalytic site of the kinase. Presumably the ~20 intervening residues would be more than sufficient to span the distance, but it would be nice to have this explicitly mentioned.

To address this point we generated a model of TACC3 bound to phosphorylated Aurora-A using FlexPepDock and include analysis of this in the revised version (Figure 3G) “This conformation is incompatible with binding of peptide substrates to the active site, which explains why the region of TACC3 around S558 is not bound to the active site. This observation raises the question of whether a single molecule of TACC3 could span both the docking site 1 and the substrate binding site of Aurora-A. To address this point, we generated a model of TACC3 bound to active Aurora-A using FlexPepDock Server (London et al, 2011; Raveh et al, 2010) and the structure of an active Aurora-A, phosphorylated on Thr288 in complex with TPX2 (Fig. 3G). The surface of Aurora-A in the vicinity of the TACC3 site-1 is virtually identical between active and inactive states of the kinase, and the TACC3 docking peptide, aa522-537 adopted a stable conformation, similar to that found in the crystal structure. We modeled the interaction of the substrate region of TACC3 (aa553-561) based on the structure of a substrate-like inhibitor bound to protein kinase A (PKA). The TACC3 sequence was readily accommodated into the equivalent binding pocket of Aurora-A. The 16 residues gap between the docking and substrate motifs is more than sufficient to span the distance of 24 Å between their binding sites on Aurora-A because an extended peptide has a spacing of 3.5 Å per residue. “

3) It is unclear whether the kinase activity assays showing activation of AurA by TACC3 (Figure

S2) were done with phosphorylated or unphosphorylated AurA. If activation of AurA by TACC3 requires activation loop phosphorylation (unlike activation by Tpx2) then this might help explain the inactive conformation of AurA observed in the x-ray structure (point 1 above). If so, perhaps the authors could make this point more explicit.

The kinase assays in Fig EV2C were done using phosphorylated Aurora-A. This has been clarified in the text. The new panel in Fig 2 was done with unphosphorylated Aurora-A and this is stated in the legend and methods. In combination, the results show TACC3 activation of Aurora-A is not dependent on the phosphorylation status of Aurora-A.

4) The authors state that the AurA/TACC3 interaction is conserved in drosophila, but do not comment on whether the interface visualized in their x-ray structure is also conserved across metazoans. Admittedly this may be hard to address given the high degree of sequence conservation in metazoan kinases.

To address this comment, we have produced a sequence alignment of the interaction regions in TACC3 and Aurora-A for a selection of orthologs (Fig EV2D). The key interaction residues are largely conserved in both proteins.

5) Site 2 in the AurA:TACC3 structure looks superficially like the AurA:Tpx2 interaction. Can the authors comment on this if there is indeed any real resemblance?

There is an overlay in binding site, which we have noted in the revised manuscript. There is no conserved secondary structure between TPX2 and TACC3 (This region of TPX2 is unfolded, while the TACC3 is helical) and the chains run in opposite directions. However, there is a conserved Phe-Phe stacking interaction between Aurora-A Phe157 and TPX2 Phe19/TACC3 Phe549.

6) It is not clear why the R555A/K556A double mutant was tested for disruption of the TACC3:CHC interaction as opposed to the single R555A or double R555A/K562A mutants. Can they clarify this point?

The single R555A mutant peptide has a similar affinity for CHC as the R555A/K562A or unphosphorylated WT peptide (data included in revised Figure 7). This is consistent with the model in which R555 forms a salt-bridge with pS558.

7) the crystallographic data table lists an Rfree value that is lower than the Rwork value for the CHC/TACC3 structure. This is probably a typo, as lower Rfree values are very rare and usually indicate incorrect assignment of the test set flags.

This was a typo. We thank the reviewer for their observation and have corrected the table.

8) the authors cite Filappakopoulos et al 2008 for the Abl:SH2 NCAP interaction, but this was technically first shown by the Kuriyan group (Nagar et al Mol. Cell 2006). The authors also omit several other relevant references for the Tpx2 activation mechanism cited in the introduction, e.g. Zorba et al 2014, Cyphers et al., 2017.

We have updated our references to include these omissions.

2nd Editorial Decision

24 January 2018

Thank you for submitting your revised manuscript for our consideration. We have now heard back from the three original referees, and I am pleased to inform you that they all consider the manuscript significantly improved and therefore in principle suitable for EMBO Journal publication. The only issue that I feel still needs to be addressed prior to ultimate acceptance concerns the results in Figure 6 D and their interpretation (see the comments of referee 2) - this would require either providing additional control data such as requested by the referee, or alternatively toning down the conclusions from these experiments.

When re-revising the manuscript, please also take care of the following important editorial points:

- Microscopy scale bars are very heterogeneous throughout the figures, and may in some cases (e.g. Fig EV4) be too small/thin to be visible in the final type-set version - therefore, please revisit and, where necessary, redraw the scale bars in all figures, referring to our Figure Preparation guide (http://embopress.org/sites/default/files/EMBOPress_Figure_Guidelines_061115.pdf) for guidance.
- The format of the reference list (journal abbreviations, number of authors listed) needs to be consistently adjusted to EMBO Journal style.
- Please include a dedicated "Data Availability" section at the end of the Material and Methods (suggested wording: "The [structural coordinates | microarray | mass spectrometry] data from this publication have been deposited to the [name of the database] database [URL] and assigned the identifier [accession | permalink | hashtag].")
- Pre-acceptance checks by our data editors have raised several queries regarding data descriptors in the figure legends.
- Finally, please include suggestions for 2-5 one-sentence 'bullet points', containing brief factual statements that summarize key aspects of the paper - they will form the basis of an editor-written 'Synopsis' accompanying the online version of the article. Please see the latest research articles on our website (emboj.embopress.org) for examples. The synopsis also includes a simplified schematic image (550 pixels wide and max 400 pixels high), but unless you want to provide a specific drawing for this purpose, I would simply adopt the summary scheme already present in Figure 8 of the article.

=> Attached, please find an edited/commented Word document with activated "Track changes" option. I would appreciate if you incorporated the requested final text modifications and answered the Figure legend queries directly in this version, uploading the edited main text document upon resubmission with changes/additions still highlighted via the "Track changes" option.

Once we will have received the re-revised manuscript files addressing the remaining editorial and referee points, we should hopefully be in a position to proceed with eventual acceptance and publication of the study. Please do not hesitate to get back to me should you have any further questions in this regard.

REFEREE REPORTS

Referee #1:

The authors have carefully considered the reviewer concerns and have thoughtfully added additional analysis and data that address concerns. The data demonstrating the enhancement of AuroraA autocatalytic activity upon TACC3 binding (Fig 2E) and that mutations to the CHC surface inhibit TACC3 association (Fig. 6B) are particularly welcome. The analysis of the allosteric mechanism of AuroraA activation also adds positively to that part of the study. The manuscript has been improved greatly and is suitable for publication. One last comment/suggestion: considering the relatively poor quality of the x-ray diffraction data for the CHC-TACC3 structure and the electron density map into which the peptide was built, I strongly encourage inclusion of the details in Appendix Figure 3 as an Expanded View Figure that readers can access. The details were critical for having confidence that the model was built accurately.

Referee #2:

The authors have adequately responded to the majority of my concerns/criticisms, and have added data that improves the manuscript.

I am still concerned that the experiment and interpretation for Fig 6D is flawed, and not sure how much of a conclusion can be taken away here. The authors want to argue that mutations in the

predicted CHC-TACC3 interface on CHC should abolish CHC recruitment to spindles, via reduced ability to interact with TACC3 and create a MT-binding interface. The data, however, would suggest that even the WT CHC-GFP construct used is unable to interact TACC3 properly, as it is not able to bring TACC3 back to the spindle, and thus the CHC-GFP localization here is independent of TACC3. The authors did not try to address the original concern of this failure to recruit TACC3 also brought up by another referee. Can they rule out that the mutations are non-specifically destabilizing the CHC-GFP? What do protein levels look like here? Could the authors support their assumptions by showing that CHC-GFP transgene actually interacts with TACC3 in mitotic cells (i.e. via IP) and the mutants do not (or less)? Another explanation may be that the amount of CHC-GFP in these cells could simply not be enough to support interaction/recruitment of TACC3, and thus not appropriate as a biological readout of their interaction. We don't know how these levels compare to endogenous CHC levels. Perhaps a western showing bulk levels of CHC and CHC-GFP?

As I stated before, I think the strength and novelty of this work lies in the biochemical/structural data/hypotheses. I don't think Fig 6D is critical for the overall work, but also don't think conclusions may be drawn from this in the current form.

Referee #3:

Burgess et al have systematically addressed all the key concerns of the reviewers and added several new sets of experiments that nicely support their story. The new NMR data in particular are a big improvement and much more strongly support the proposed helical structure of the phosphorylated TACC3 peptide. I am also glad to see that the authors now propose a mechanism by which TACC3 binding activates AurA, and although this mechanism is somewhat speculative, its inclusion nonetheless makes the story more compelling.

Overall this is a nice paper, and is suitable for publication in EMBO J.

2nd Revision - authors' response

1 February 2018

We have revised the manuscript as you requested, and we hope that it is now acceptable for publication in *EMBO Journal*. We have provided a separate response to the reviewer comment on the results in Figure 6D. Your previous letter is included below, along with our response to the individual points.

We thank you and the reviewers for helping us to improve our manuscript.

Thank you for submitting your revised manuscript for our consideration. We have now heard back from the three original referees, and I am pleased to inform you that they all consider the manuscript significantly improved and therefore in principle suitable for EMBO Journal publication. The only issue that I feel still needs to be addressed prior to ultimate acceptance concerns the results in Figure 6 D and their interpretation (see the comments of referee 2) - this would require either providing additional control data such as requested by the referee, or alternatively toning down the conclusions from these experiments.

We toned down the conclusions from these experiments – see response to Reviewers.

When re-revising the manuscript, please also take care of the following important editorial points:

- Microscopy scale bars are very heterogeneous throughout the figures, and may in some cases (e.g. Fig EV4) be too small/thin to be visible in the final type-set version - therefore, please revisit and, where necessary, redraw the scale bars in all figures, referring to our Figure Preparation guide (http://embopress.org/sites/default/files/EMBOPress_Figure_Guidelines_061115.pdf) for guidance. *Microscopy scale bars have been adjusted in line with requirements.*

- The format of the reference list (journal abbreviations, number of authors listed) needs to be

consistently adjusted to EMBO Journal style.

The references are reformatted in line with EMBO Journal style.

- Please include a dedicated "Data Availability" section at the end of the Material and Methods (suggested wording: "The [structural coordinates | microarray | mass spectrometry] data from this publication have been deposited to the [name of the database] database [URL] and assigned the identifier [accession | permalink | hashtag].")

We have included a DATA AVAILABILITY section as requested.

- Pre-acceptance checks by our data editors have raised several queries regarding data descriptors in the figure legends.

We have addressed all queries.

- Finally, please include suggestions for 2-5 one-sentence 'bullet points', containing brief factual statements that summarize key aspects of the paper - they will form the basis of an editor-written 'Synopsis' accompanying the online version of the article. Please see the latest research articles on our website (emboj.embopress.org) for examples. The synopsis also includes a simplified schematic image (550 pixels wide and max 400 pixels high), but unless you want to provide a specific drawing for this purpose, I would simply adopt the summary scheme already present in Figure 8 of the article.

We have included 4 bullet points to form the basis of the 'synopsis', and we agree that you should adopt Figure 8 for this purpose.

=> Attached, please find an edited/commented Word document with activated "Track changes" option. I would appreciate if you incorporated the requested final text modifications and answered the Figure legend queries directly in this version, uploading the edited main text document upon resubmission with changes/additions still highlighted via the "Track changes" option.

We followed your instructions and so our changes are highlighted via Track Changes.

We thank the reviewers for their careful reading of our work and their thoughtful feedback.

Referee #1:

I strongly encourage inclusion of the details in Appendix Figure 3 as an Expanded View Figure that readers can access. The details were critical for having confidence that the model was built accurately.

The Appendix Figures will be made available online and are as easy to access for readers as the Expanded View. Therefore, because we were short of space in EV, we decided to leave the figure in the Appendix.

Referee #2:

I am still concerned that the experiment and interpretation for Fig 6D is flawed, and not sure how much of a conclusion can be taken away here. The authors want to argue that mutations in the predicted CHC-TACC3 interface on CHC should abolish CHC recruitment to spindles, via reduced ability to interact with TACC3 and create a MT-binding interface. The data, however, would suggest that even the WT CHC-GFP construct used is unable to interact TACC3 properly, as it is not able to bring TACC3 back to the spindle, and thus the CHC-GFP localization here is independent of TACC3. The authors did not try to address the original concern of this failure to recruit TACC3 also brought up by another referee. Can they rule out that the mutations are non-specifically destabilizing the CHC-GFP? What do protein levels look like here? Could the authors support their assumptions by showing the that CHC-GFP transgene actually interacts with TACC3 in mitotic cells (i.e. via IP) and the mutants do not (or less)? Another explanation may that the amount of CHC-GFP in these cells could simply not be enough to support interaction/recruitment of TACC3, and thus not appropriate as a biological readout of their interaction. We don't know how these levels compare to endogenous CHC levels. Perhaps a western showing bulk levels of CHC and CHC-GFP?

We agree with the reviewer and their concerns are now stated in the revised manuscript. We are also now more cautious in our interpretation. The new paragraph, with changes underlined reads:

We then re-expressed the designated GFP-CHC constructs in HeLa cells depleted of endogenous CHC. In line with previous experiments, WT GFP-CHC, but not GFP alone showed enrichment at the spindle, although there is a background of cytoplasmic CHC (Fig 6D). All CHC mutants tested were impaired in spindle localization: a control in which the TACC3 interaction site is deleted (Δ 457-507); mutations in the TACC3 binding site (L476D, L480D; R481E; L503A, Y504A) and mutation of a residue that holds R481 in place (D455K). However, these data should be interpreted cautiously because rescue of CHC knockdown with exogenous WT-CHC was unable to fully restore TACC3 localisation to the spindle. Either the exogenously expressed CHC is impaired in binding TACC3 or the levels of exogenous CHC are too low, and so we cannot draw definitive conclusions from the rescue experiments with CHC mutants. Moreover, the results suggest that the localisation of CHC to the spindle has a component that is independent of the interaction we have characterised, most likely the N-terminus of clathrin/TACC domain of TACC3, and other components of the complex such as GTSE1 (Hubner et al. 2010; Hood et al., 2013).

3rd Editorial Decision

2 February 2018

Thank you for submitting your final revised manuscript for our consideration. I am pleased to inform you that we have now accepted it for publication in The EMBO Journal.

Corresponding Author Name: Richard Bayliss

Manuscript Number: EMBOJ-2017-97902